# Amphotericin B resistance in *Leishmania mexicana*: Alterations to sterol metabolism and oxidative stress response

Edubiel A. Alpizar-Sosa[1,2], Nur Raihana Binti Ithnin[1,3], Wenbin Wei[2], Andrew W. Pountain[1,4], Stefan K. Weidt[5], Anne M. Donachie[1], Ryan Ritchie[1], Emily A. Dickie[1,5], Richard J. S. Burchmore[1,5], Paul W. Denny[2], Michael P. Barrett[1,5]*

1 Wellcome Centre for Integrative Parasitology, School of Infection & Immunity, College of Medical, Veterinary and Life Sciences, University of Glasgow, Glasgow, United Kingdom, 2 Department of Biosciences, Durham University, Durham, United Kingdom, 3 Department of Medical Microbiology, Faculty of Medicine & Health Sciences, Universiti Putra Malaysia, Serdang, Selangor, Malaysia, 4 Institute for Computational Medicine, New York University Grossman School of Medicine, New York City, New York, United States of America, 5 Glasgow Polyomics, College of Medical, Veterinary & Life Sciences, University of Glasgow, Garscube Estate, Bearsden, Glasgow, United Kingdom

* michael.barrett@glasgow.ac.uk

**Data Availability Statement:** For sequencing (WGS), all high throughput sequencing data generated in these experiments were submitted to

## Abstract

Amphotericin B is increasingly used in treatment of leishmaniasis. Here, fourteen independent lines of *Leishmania mexicana* and one *L. infantum* line were selected for resistance to either amphotericin B or the related polyene antimicrobial, nystatin. Sterol profiling revealed that, in each resistant line, the predominant wild-type sterol, ergosta-5,7,24-trienol, was replaced by other sterol intermediates. Broadly, two different profiles emerged among the resistant lines. Whole genome sequencing then showed that these distinct profiles were due either to mutations in the sterol methyl transferase (C24SMT) gene locus or the sterol C5 desaturase (C5DS) gene. In three lines an additional deletion of the miltefosine transporter gene was found. Differences in sensitivity to amphotericin B were apparent, depending on whether cells were grown in HOMEM, supplemented with foetal bovine serum, or a serum free defined medium (DM). Metabolomic analysis after exposure to AmB showed that a large increase in glucose flux via the pentose phosphate pathway preceded cell death in cells sustained in HOMEM but not DM, indicating the oxidative stress was more significantly induced under HOMEM conditions. Several of the lines were tested for their ability to infect macrophages and replicate as amastigote forms, alongside their ability to establish infections in mice. While several AmB resistant lines showed reduced virulence, at least two lines displayed heightened virulence in mice whilst retaining their resistance phenotype, emphasising the risks of resistance emerging to this critical drug.

## Author summary

Amphotericin B is increasingly used in treatment of leishmaniasis. Antimicrobial resistance is a persistent threat for any drugs and although reports of resistance to

the NCBI National Center for Biotechnology Information (https://www.ncbi.nlm.nih.gov) repository project PRJNA763929 and PRJNA770503. For LCMS all raw files data generated were submitted to the Metabolomics repository MetaboLights (https://www.ebi.ac.uk/metabolights/) with project number MTBLS2744. All other relevant data and accompanying Supporting Information files are provided within this manuscript.

**Funding:** EAAS was funded by a CNPq (National Council for Scientific and Technological Development - Brazil, Scholarship 201535/2015-7). NRI was funded by Ministry of Higher Education of Malaysia and Universiti Putra Malaysia (KPT (BS) 850101016636). EAAS and PWD were funded by MRC Confidence in Concept MC-PC-17157 and UKRI Grand Challenges Research Fund ("A Global Network for Neglected Tropical Diseases") grant number MR/P027989/1. MPB was funded by an MRC Newton grant (MR/S019650/1 -"Bridging epigenetics, metabolism and cell cycle in pathogenic trypanosomatids") and is also part of the core funded Wellcome Centre for Integrative Parasitology (grant number 104111/Z/14/Z). The funders had no role in study design, data collection and analysis, decision to publish, or preparation of the manuscript.

**Competing interests:** The authors have declared that no competing interests exist.

amphotericin B are rare in the field, it is something that cannot be ignored since this drug is used in leishmaniasis relapse cases and is indicated to treat other infectious diseases such as mycoses. Here, we report the selection and characterisation of fourteen independent lines of *Leishmania mexicana* and one of *L. infantum* resistant to amphotericin B or its analogue nystatin. Sterol profiles were altered in all resistant lines compared to wild-type. The predominant wild-type sterol, ergosta-5,7,24-trienol, was replaced by other sterol intermediates. In one set of mutants this could be attributed to loss of heterozygosity derived from mutations in the sterol methyl transferase (C24SMT) gene locus and in a second set the sterol C5 desaturase (C5DS) gene was changed. Differences in sensitivity to amphotericin B were apparent in cells grown in serum free defined medium compared to serum containing HOMEM and metabolomics analysis indicated that AmB promotes oxidative stress when serum is present but not in its absence. While some lines lost virulence in mouse infections others displayed heightened virulence in mice whilst all cell lines retaining their resistance phenotype emphasising the risk of resistance emerging in patients.

## Introduction

The Leishmaniases are caused by parasitic protozoa of the genus *Leishmania* and are transmitted via the bite of infected female phlebotomine sand flies. Clinical forms range from self-healing but disfiguring cutaneous (CL) and mucocutaneous forms (MCL) [1–3] to potentially fatal visceral leishmaniasis (VL) [2], which can also lead to post-kala-azar dermal leishmaniasis (PKDL) in some instances following treatment [4]. No vaccine against human leishmaniasis exists [5], and patient management relies on chemotherapy, which is currently unsatisfactory for reasons including cost, toxicity [6,7], duration and mode of administration and drug resistance [1,8–10]. Amphotericin B (AmB) is a natural product used to treat leishmaniasis [11,12]. It is a polyene antimicrobial and was initially introduced for anti-fungal use alongside other polyenes, such as nystatin (Nys) [13–21]. The introduction of liposomal formulations of AmB, AmBisome, significantly reduced inherent toxicity [12,22–24] and has become the preferred treatment for VL [8,25,26], especially while costs have been kept low due to donations of the drug to the WHO for use in countries most affected by the disease [27–32]. AmB is used in the elimination program of VL in south Asia [1] and has remained effective in VL patients refractory to other antileishmanials [33]. It is also active against other forms of the disease [34,35], although it has been used less for CL [12,36]. The availability of donated AmBisome has enabled a campaign of single dose use of the drug against VL. Treating large numbers of individuals in this way avoids complications associated with multiple-dosing regimens and has been of public health benefit [37–39], contributing to a decline in infections as part of a WHO led campaign to eliminate VL as a public health problem [40,41]. That campaign, however, has assumed that the development of resistance to AmB in *Leishmania* parasites is not a significant problem. This is based on the widely held perception that, despite over 70 years of clinical use as an anti-fungal agent, this use has not been accompanied by a significant emergence of AmB resistance (AmBR) in fungal species [42–49]. AmB, and other antimicrobial polyenes, bind to ergostane-based membrane sterols with high avidity, and ergosterol and related sterols are the predominant forms in fungi and in *Leishmania* spp. [16,50–54]. The compounds bind to cholesterol, the major sterol in mammalian cell membranes, with markedly less avidity [55]. This selectivity is driven by the altered structural configuration of ergosterol and its analogues relative to cholesterol: differences include double bonds at positions C7 and C22, and a methylation at C24 of the side chain [56–58].

In spite of the perception that AmBR is only a minor risk, several examples of resistance [59–64] and treatment failure with AmB [47,65–67] have been described in leishmaniasis cases, and polyene resistance is by no means unknown in fungal disease [49,63,68–78]. An outbreak of autochthonous canine VL in Uruguay found clinical isolates 3 to 4-fold less susceptible to AmB [79,80]. Similar differences in susceptibility to AmB were found in a non-endemic region in India [62] and patients with cutaneous leishmaniasis in Colombia [81]. Development of resistance to AmB is a major threat for the treatment of VL [82]. Mechanisms of resistance appear to be diverse. For example, fungal species with a normal content of ergosterol and high levels of catalase are intrinsically resistant to AmB [45,83,84]. Likewise, in *Leishmania* spp., numerous enzymes involved in protection against reactive oxygen species (ROS) appear to associate with AmBR. These include the polyamine-trypanothione pathway (PTP) [59,85,86], L-asparaginase [87], cysteine synthase [88], ascorbate peroxidase [89], tryparedoxin peroxidase [90] and the silent information regulator 2 (Sir2) protein [82]. Furthermore, cross resistance to AmB is sometimes observed between lines resistant to the antileishmanials miltefosine and antimonials [13,64,91–94], reflecting a multifactorial mode of action (MoA) of AmB [43,59]. Although AmB is used preferentially against visceral leishmaniasis caused by *L. donovani* or *L. infantum*, rather than cutaneous leishmaniasis caused by *L. mexicana*, the high degree of similarity between *Leishmania* species often enables observations in one organism to allow inference in others. The relative simplicity of working with *L. mexicana* makes it a popular model for obtaining knowledge that might apply to other *Leishmania* species more widely.

We have previously shown that the loss of the wild type ergostane-type sterol relates to selection of AmBR in laboratory-generated mutants of *Leishmania mexicana*. Genome sequence analysis of resistant mutants has shown that mutations in genes encoding lanosterol 14-alpha demethylase (C14DM) [95], C5-sterol desaturase (C5DS) and C-24 sterol methyltransferase (C24SMT) [96] can all lead to AmBR. Additionally, sterol profiling revealed loss of the wild type ergosterol or its isomers (ergosta-5,7,24-trienol also known as 5-dehydroepisterol in *L. mexicana*), with inferred loss of drug binding in each case. Other studies in AmBR *Leishmania* spp. have also shown loss of expression of one of the two C24SMT-transcripts [59,97–99] and structural variations at this locus, alongside loss of function of the miltefosine transporter in some instances [91,96]. Moreover, using *L. major* and *L. donovani*, a series of gene knockout studies have also shown that loss of C14DM [100–102], C24SMT [99,103] and C5DS [104] all lead to the creation of AmBR-parasites that are viable *in vitro*, albeit of varying degrees of fitness. Here we describe the characterisation of fifteen new independently selected polyene-resistant lines, fourteen of *L. mexicana* (ten selected to AmB and four to Nys), plus one AmBR-*L. infantum* line. In each case sterol profiling pointed to a loss of wild type ergosterol and its analogue (ergosta-5,7,24-trienol) in *Leishmania* with essentially two different types of compensatory sterol-profile arising, which are each accounted for by mutations to mutations in C24SMT or C5DS. We also assessed the fitness of the selected lines in macrophages and in mice and the stability of the resistance phenotype. Since studies so far have indicated that mutations to different genes in the sterol pathway can lead to AmBR [95,96], and other instances of resistance showed no apparent change to sterol metabolism [86] we anticipate that by expanding the numbers of resistant lines selected could amplify our knowledge of the different mechanisms capable of yielding AmBR.

## Results

### Polyene resistance selection in fifteen independent *Leishmania* spp. lines

A total of fifteen independent lines of *Leishmania* spp. (S1 Table) were selected for polyene resistance via a stepwise growth in increasing concentration (S1 Fig) of AmB or Nys added to

the culture media HOMEM or a serum-free defined medium (DM) (S2 Table). Seven AmB resistant *L. mexicana* lines (AmBRcl.14, AmBRcl.8, AmBRcl.6, AmBRcl.3, AmBRA4, AmBRB2 and AmBRC1) were selected in complete HOMEM and three lines (AmBRA3-DM, AmBRB2-DM and AmBRC2-DM) were selected in DM. In addition, four Nys resistant (NysR) *L. mexicana* lines (NysRcl.B2 NysRcl.C1, NysRcl.E1, and NysRcl.F2) and one AmBR *L. infantum* (AmBRcl.G5) were also selected in complete HOMEM. The levels of resistance attained in *L. mexicana* after AmB exposure for 210 days in HOMEM were between 4.5- and 12.8-fold compared with the wild type cultured in parallel in the absence of drug. A comparable increase in resistance was observed in all four lines exposed to Nys for 140 days ranging from 6.6- to 11-fold and to a lesser extent in AmBRcl.G5 (*L. infantum*) which increased by 3.7-fold after drug pressure for 105 days (Fig 1A–1C). In contrast, resistance attained in all three AmBR *L. mexicana* lines selected in DM over 372 days was only 2.2-fold that of the wild type in DM (Fig 1D). Levels of resistance to AmB or Nys were stable after at least 10 passages in the absence of drug and confirmed in at least two clonal populations derived by limiting dilution from each of the independent fourteen polyene resistant *L. mexicana* lines (for AmBR-*L. infantum*, only one clonal population could be recovered and analysed). AmB can be sequestered by cholesterol and lipoproteins present in the serum [105,106]. Surprisingly, wild type cells sustained in serum-free conditions (DM) were 3.6-fold less susceptible to AmB ($EC_{50}$ 217.3) than they were in HOMEM ($EC_{50}$ 60 nM) indicating an enhanced overall potency of the drug against parasites in rich medium. Parasite growth rate (S2A–S2C Fig), final cell density (i.e., stationary phase) and cell body length (S2D Fig) of AmBR and NysR resistant lines cultured in HOMEM were comparable to the parental line cultured in parallel without drug whilst in DM AmBR resistant lines the growth rate was slightly lower than in wild type.

## Response of polyene resistant lines to other antileishmanials

Susceptibility to other antileishmanials was measured in only one resistant clonal population derived from each of the fifteen lines derived by limiting dilution (S1 Table). Considering the similar mode of action between polyenes, reciprocal cross-resistance between AmBR lines to Nys and *vice versa* was expected. AmBR lines were 12.9 to 25-fold cross-resistant to Nys and to a lesser extent (1.8 to 2.0-fold), to the smaller polyene natamycin (NMC). Similarly, NysR lines cross-resistance to AmB was from 5.8 to 9.1-fold. Interestingly, variable cross-resistance to the antileishmanial miltefosine (MF) was found across all polyene resistant lines with the lowest change found in lines sustained in DM (1.1 to 1.4-fold) while cells selected in HOMEM showed changes ranging from 2.3 to 5.3-fold increases in most lines, rising to 10-fold in two lines, AmBRcl.8 and AmBRcl.6, in which we found a deletion of the miltefosine transporter (described in a later section) (Fig 1A). This locus has occasionally been associated with cross resistance between AmB and MF in previous studies [91,93,94,96]. A range of changes in $EC_{50}$ were obtained with two inhibitors of lanosterol 14-alpha demethylase (C14DM), ketoconazole and fenarimol [100,107–110], and to the antileishmanial drug, potassium antimony tartrate (PAT). All fourteen polyene-resistant *L. mexicana* lines cultured in HOMEM and DM were significantly more susceptible to paromomycin (PAR) (P = 0.0001 to 0.0319), with a decrease of 1.7 to 10-fold in the $EC_{50}$ irrespective of the culture medium (Fig 1 and S3 Table). Similarly, increased susceptibility to pentamidine (PENT) and to methylene blue (MB) was statistically significant in all polyene resistant lines sustained in HOMEM but not in DM.

$EC_{50}$ value to PENT decreased by 7- to 10.9–fold (P = 0.0008 to P<0.0001) in all seven AmBR lines and from 2- to 3.8-fold (P = 0.01 to P<0.0001) in all four NysR lines, selected in HOMEM (Fig 1 and S3 Table). Recently, Mukherjee et al. related changes in sterol composition to altered mitochondrial membrane potential ($\Delta\psi M$) which they implicated in

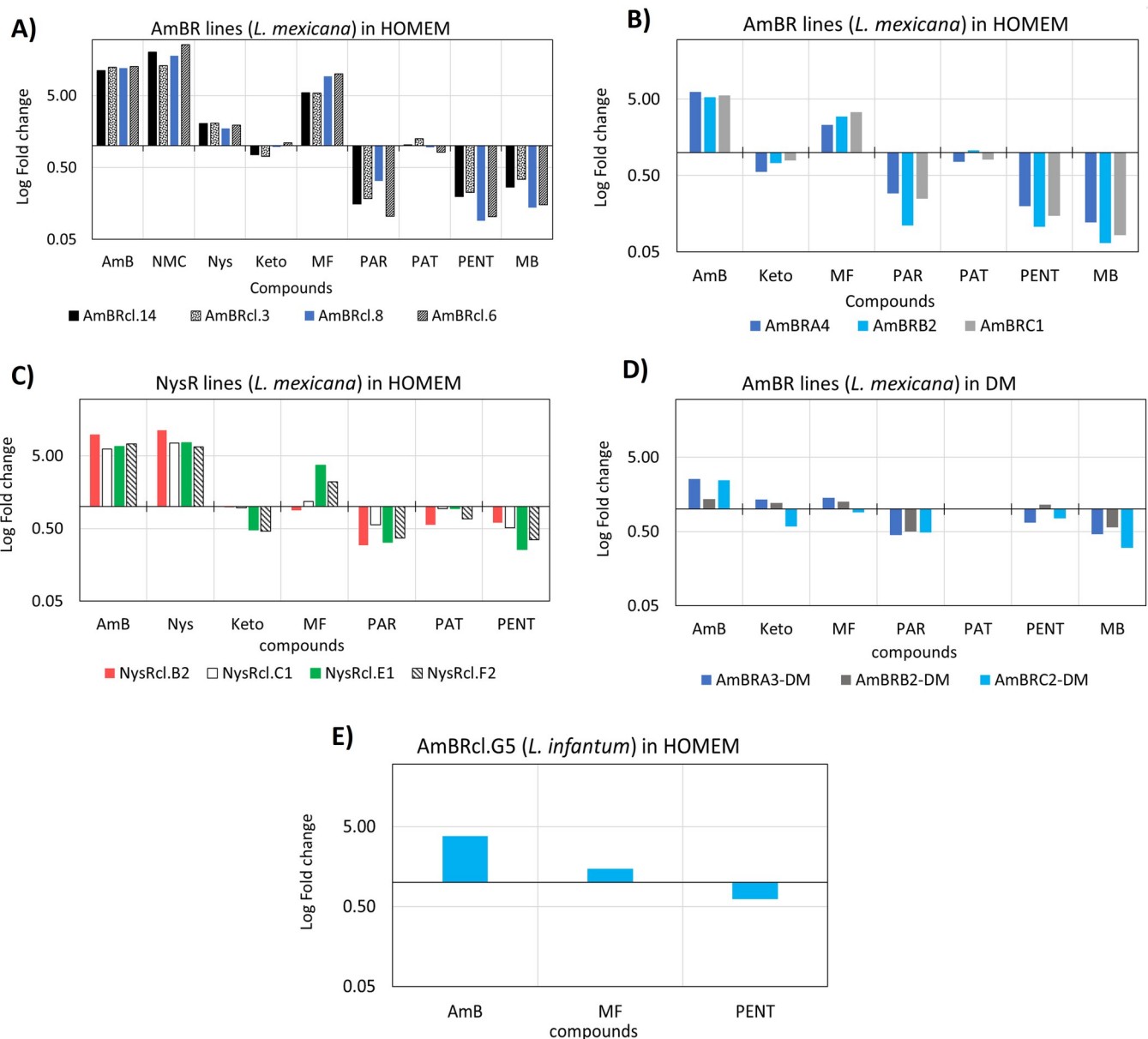

**Fig 1. Fold change in EC$_{50}$ in all polyene resistant lines of *Leishmania spp*.** Log Fold change (EC$_{50}$) in resistant lines is relative to the respective parental wild type cultured in parallel without drug. Fig 1A and 1B) AmBR-*L. mexicana* in HOMEM, Fig 1C) NysR-*L. mexicana* in HOMEM, Fig 1D) AmBR- *L. mexicana* in DM and E) AmBR-*L. infantum* in HOMEM. Values higher and lower than 1, indicate a decrease and rise in susceptibility relative to the parental line, respectively. A complete list of values in fold-change and EC$_{50}$ with statistical analysis is provided in S3 Table. AmB (amphotericin B), Nys (nystatin), Keto (ketoconazole), MF (miltefosine), PAR (paromomycin), PAT (potassium antimony tartrate), PENT (pentamidine) and MB (methylene blue).

hypersensitivity to PENT whose uptake into mitochondria depends on ΔψM [102]. To a lesser extent, AmBR lines grown in DM also revealed increased susceptibility to PENT (P = 0.0938 and P = 0.2143), except AmBRB2 which was marginally less sensitive (1.15-fold) to the drug. It is possible the differences in oxidative stress induction by AmB in different media also relate to this difference in PENT sensitivity between resistant lines selected in different media. A comparable rise in susceptibility to PENT (P ≤ 0.05) was also observed in clemastine fumarate-resistant *L. major* promastigotes with altered membrane lipid composition [111], further

indicating potential trade-offs between AmB and PENT resistance mechanisms, centred around lipid production. Exposure to MB, an oxidative stress inducing agent [112], showed that the $EC_{50}$ of MB in AmBR lines grown in HOMEM was 2.9 to 15.3-fold lower than in wild type cells (P = 0.0001 to 0.0084) (S3 Table), MB induces intracellular oxidative stress by increasing the NADP+: NADPH ratio and stimulating the oxidation of key cellular redox related thiols such as glutathione [113,114] the functions of which are replaced by trypanothione in trypanosomatids. To a lesser extent (1.76 to 3.34-fold), hyper-susceptibility to MB was also found in AmBR lines selected in DM (AmBRA4, P = 0.506; AmBRB2, P = 0.114 and AmBRC1, P = 0.012) (Fig 1D and S3 Table). It has been reported that a range of *Candida* species in which AmBR arose through changes to sterol metabolism had increased in sensitivity to oxidative stress [76], and the data here point to similar associations in *Leishmania*.

## Metabolic response to AmB exposure

AmB kills *Leishmania* very quickly. At high concentration of AmB (i.e., 5 x the $EC_{50}$) death ensues within 30 min. We used untargeted metabolomics to determine what changes occur to cellular metabolism over 15 mins by exposing parasites to high concentrations of AmB equivalent to 5x their respective AmB-$EC_{50}$ values in HOMEM: 0.31 µM (parents LmWT1 and LmWT2); 3.0 µM (AmBRcl.14 and AmBRcl.8) and 1.74 µM (AmBRB2). Likewise, cell lines cultured in DM were subjected to AmB at high concentrations (i.e., 5 x $EC_{50}$): 1.16 µM (LmWT-DM) and 3.58 µM (AmBRA3-DM).

In untreated cells no significant differences were seen in intracellular metabolites involved in the tricarboxylic acid cycle (TCA) or the pentose phosphate pathway (PPP) between HOMEM and DM (S3 Fig). We then compared the metabolomes of cells exposed to drug that had been cultured under the differing media conditions. The most notable difference was in the major increase in metabolites of the PPP: metabolites such as 6-phospho-D-gluconate (Adj. P-value = 0.0121 in AmBRcl.14; 0.0063 in AmBRcl.8; 0.0221 in AmBRB2 and 0.0074 in AmBRA3-DM), glyceraldehyde 3-phosphate (Adj. P-value = 0.1377 in AmBRcl.14; ND in AmBRcl.8; 0.0182 in AmBRB2 and 0.0012 in AmBRA3-DM), sedoheptulose 7-phosphate, (Adj. P-value = 0.6936 in AmBRcl.14; 0.0026 in AmBRcl.8; 0.0099 in AmBRB2 and 0.0003 in AmBRA3-DM), pentose 5-phosphates (not distinguished in this study) (Adj. P-value = <0.0001 in AmBRcl.14; ND in AmBRcl.8; 0.0016 in AmBRB2 and ND in AmBRA3-DM) and D-glucono-1,5-lactone 6-phosphate (Adj. P-value = 0.0002 in AmBRcl.8; 0.0057 in AmBRB2, not detected in other lines) were significantly increased in abundance in cells exposed to drug in HOMEM (Fig 2A) but not in DM medium (Fig 2B). Upregulation of the PPP (Fig 2C) points to a possibly increased oxidative stress response requiring rapid regeneration of NADPH [116–118]. This difference may explain why HOMEM-cultured cells are more sensitive to drug and to the oxidative stress inducer methylene blue (Fig 1 and S3 Table), with these cells experiencing heightened oxidative stress over and above that induced through cellular lysis following pore formation (triggered via AmB-ergosterol, or its isomer ergosta-5,7,24-trienol, binding). This observation may also explain why the fold-change in resistance of the lines selected in HOMEM (5.29 to 12.79-fold) is greater than those achieved in DM (1.36 to 2.54-fold) (Fig 1), as alterations in specific parts of oxidative stress response could enhance HOMEM-based resistance, although cross-resistance to MB is not seen in AmBR cells, indicating that any such response is nuanced, and the data actually indicate that in the lines selected here, decreased sensitivity to oxidative stress is not evident, indeed, as reported in *Candida spp*. [76] resistant lines showed increased sensitivity to oxidative stress.

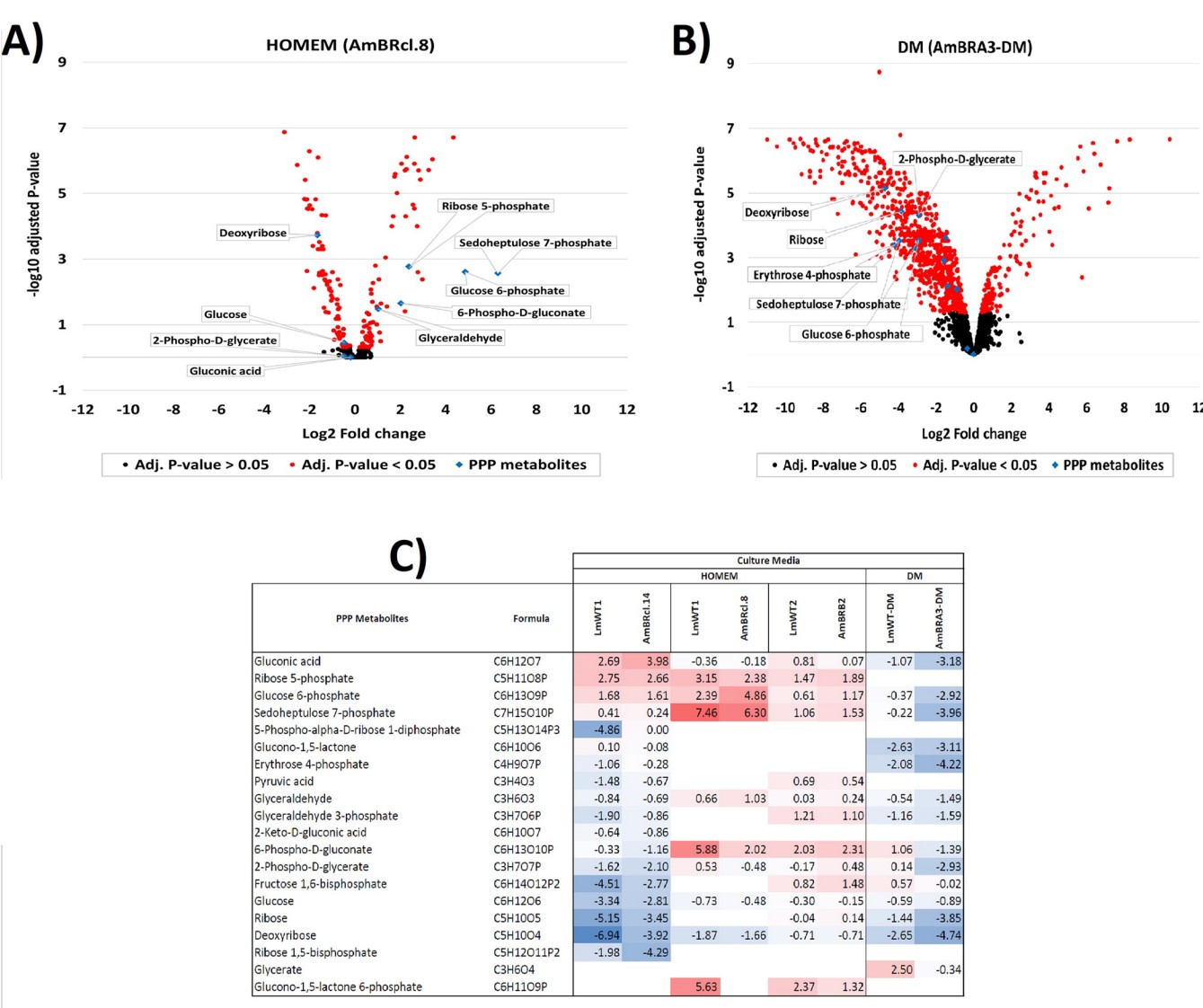

**Fig 2. Metabolic effects of exposure to AmB in *L. mexicana* promastigotes cultured in HOMEM and DM.** Mid log AmBR lines alongside with their parental wild-type *L. mexicana* promastigotes (1 x 10^8) were cultured in HOMEM (LmWT1, AmBRcl.14, AmBRcl.8, LmWT2 and AmBR2) and DM (LmWT-DM and AmBRA3-DM). Fig 2A) Volcano plot showing significant log2 fold-changes (red dots) of peaks as detected by LC-MS in promastigotes cultured in HOMEM (AmBRcl.8) and Fig 2B) in DM (AmBRA3-DM). Multivariate data analysis was performed with the PiMP analysis pipeline [115] and the Benjamini-Hochberg procedure adjusted raw P-values (q-values) < 0.05 for ANOVA. Fig 2C) Heat map of representative intermediates of the PPP. Each line represents one biological replicate (n = 4). Increase (red) or decrease (blue) in fold-change is relative to untreated cells after treatment with AmB at 5 x their $EC_{50}$ values for 15 mins. Values in bold indicate metabolites identified and all others are putative based on mass. Empty boxes (no value) indicate that the metabolite was not detected. A full list of P-values of the metabolome (including PPP intermediates) is provided in S4 Table.

## Sterol profiling in polyene resistant lines of *Leishmania* spp

Binding to ergosterol, or related ergostane-type sterols, is the primary event in the mode of action of antimicrobial polyenes [43,119]. Previous studies with AmBR *Leishmania* spp. have shown that loss of the wild type ergosterol (ergosta-5,7,24-trienol) commonly accompanies resistance development [59,96,97,120–124]. Fig 3 shows the sterol biosynthesis pathway (SBP) in *Leishmania* adapted from the KEGG database (https://www.genome.jp/kegg/) and previous work [96,123–126]. The identities and composition of the sterol peaks detected by gas

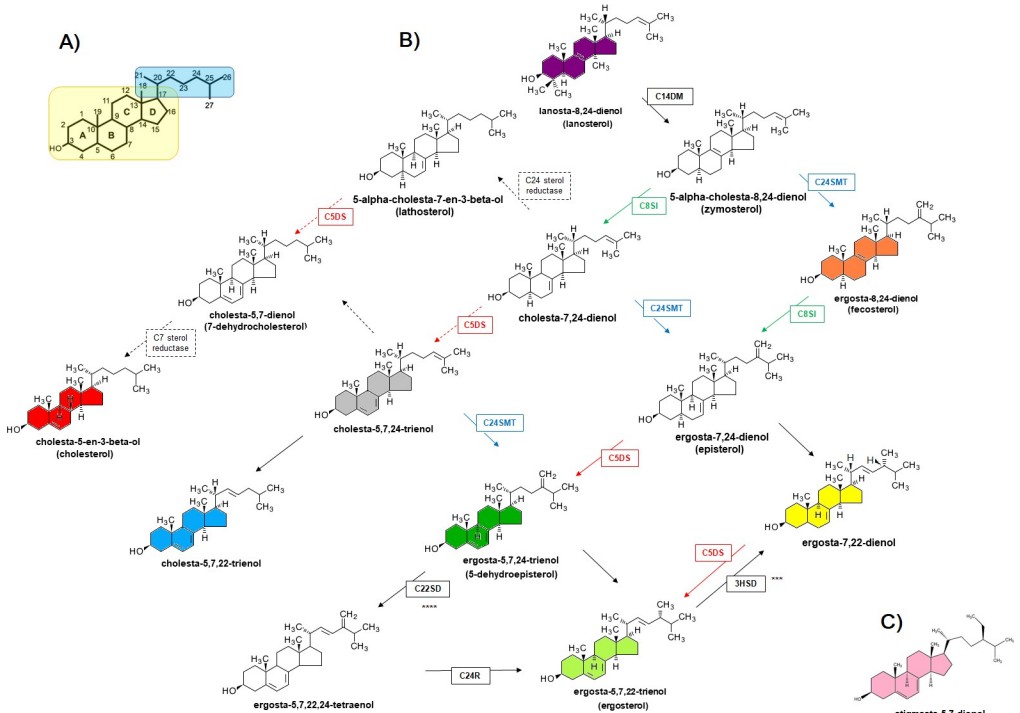

**Fig 3. Sterol Biosynthesis in *Leishmania spp.*** Fig 3A) Numbering system of the carbons in the sterol ring (boxed in yellow) and side chain (boxed in blue). Fig 3B) Molecules shadowed (colours) indicate sterols found in this work. Enzymatic steps known in mammals (dotted arrows & boxes) and in *Leishmania spp.* (solid arrows & boxes) are sterol C14 demethylase, *LmxM.11.1100* (C14DM); sterol C24-methyltransferase, *LmxM.36.2380* and *LmxM.36.2390* (C24SMT) (blue); C8-sterol isomerase, *LmxM.08_29.2140* (C8SI) (green); sterol C5 desaturase or lathosterol oxidase, *LmxM.23.1300* (C5DS) (red); sterol C22 desaturase or cytochrome p450-like protein, *LmxM.29.3550* and/or *LmxM.33.3330* (C22SD); sterol C24 reductase, *LmxM.32.0680* (C24R) and 3-beta hydroxysteroid dehydrogenase, *LmxM.18.0080* (3HSD). C) The position of stigmasta-5,7-dienol within the pathway is unknown. Pathway adapted from the KEGG Database (https://www.genome.jp/kegg/) and references [96,124****,126***].

chromatography-mass spectrometry (GC-MS) based on matches with the standards used and with the NIST library (https://www.nist.gov/srd/nist-standard-reference-database-1a-v14) are shown in S4 Fig and S5A Table.

Ergosta-5,7,24-trienol ($C_{28}H_{44}O$) was the most abundant sterol detected here in wild type *Leishmania* spp. promastigotes ranging from 72 to 82% of total sterol (Fig 4). Also known as 5-dehydroepisterol, this isomer of ergosterol differs from the major fungal ergosterol, ergosta-5,7,22-trienol [125] in the position of the double bond (carbon 24 rather than carbon 22) and in its chromatographic retention time (S5A Table) [100,126].

Another abundant sterol in wild type promastigotes was ergosta-7,22-dienol (11%) followed by cholesterol (2 to 7%) (Fig 4). Cholesterol was absent in all AmBR and wild type parasites cultured in DM (Fig 4D), as expected given that cholesterol is derived from exogenous sources, including the serum added to culture media [127,128] (and from the membrane of the host-macrophage in the case of intracellular amastigotes [129]). This likely explains its higher abundance in promastigotes (1.4 to 7.1%) and in the intracellular stage of the parasite in both wild type (12.2%) and AmBR lines (7.3 to 10.2%) cultured in HOMEM (Fig 4E and S5B Table).

All of the polyene-selected lines had altered sterol profiles compared to wild-type and these fell into two broad types. In three of the AmBR *L. mexicana* lines selected in HOMEM and all four of the Nys resistant *L. mexicana* lines, the predominant sterol became ergosta-7,22-dienol (96 to 97%) (Fig 4A and 4D), which lacks the C5 desaturation of AmB-binding ergosta-

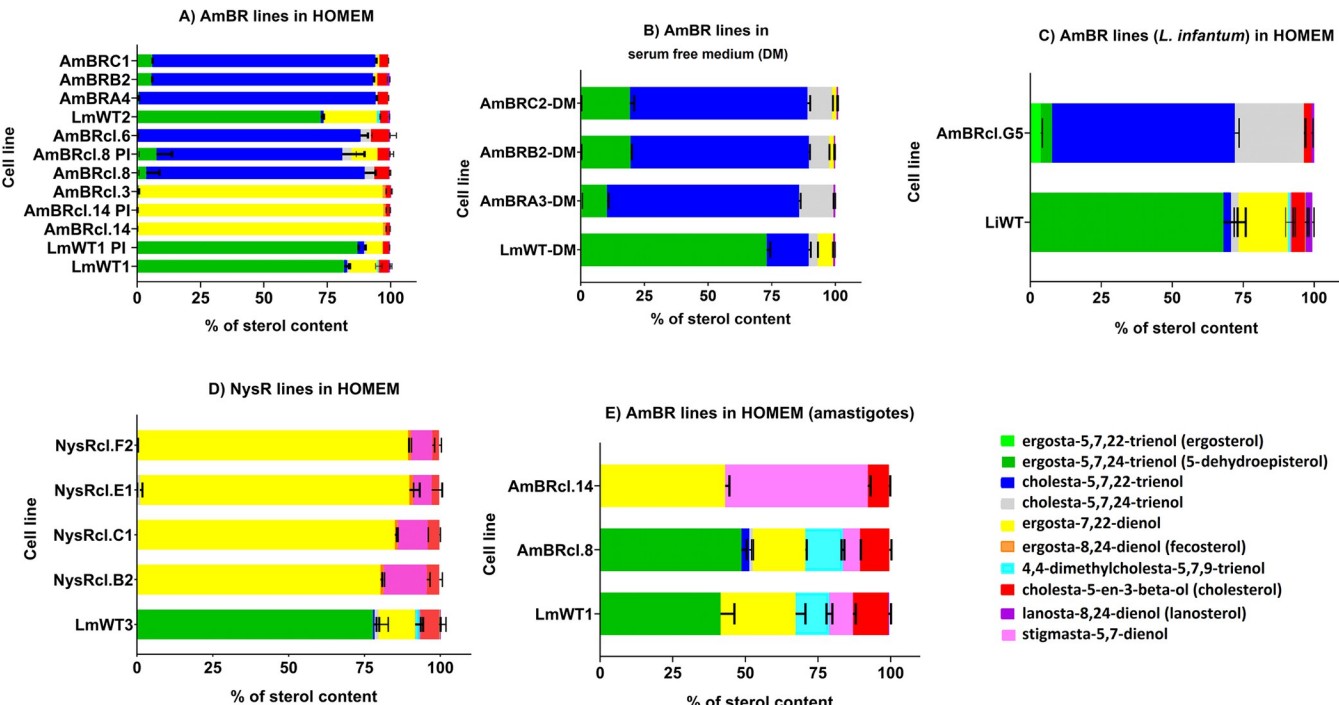

**Fig 4. Percentage of sterols (GC-MS) in polyene resistant lines of *Leishmania spp*.** Parasites were cultured as promastigotes in HOMEM (Fig 4A, 4C and 4D) and DM (Fig 4B) or as axenic amastigotes (recovered after infection in mice) in Schneider's media (Fig 4E). Species are *L. mexicana* (Fig 4A, 4B, 4D and 4E) and *L. infantum* (Fig 4C) and lines are resistant to polyenes AmB (Fig 4A, 4B, 4C and 4E) and Nys (Fig 4D). Standard deviation of the mean (bars) is from three biological replicates. See S5 Table and Material and Methods for a full description of the content and identification of sterols with GC-MS.

5,7,24-trienol. Another sterol, stigmasta-5,7-dienol, was also present in all NysR lines (NysRcl. B2 (9.9%), NysRcl.C1 (14.2%), NysRcl.E1 (6.1%) and NysRcl.F2 (7.0%)).

We also analysed the sterol content of axenic amastigotes in two resistant lines alongside their parental wild type following recovery from mouse tissue (post-infection). Parasites were maintained as amastigotes in Schneider's medium. Interestingly, stigmasta-5,7-dienol was also present in wild type *L. mexicana* axenic amastigotes (8.1% of total) and unchanged or little depleted in AmBRcl.8 (5.8%) while increased notably in AmBRcl.14 (49.1%) (Fig 4E). However, its role in resistance and its position within the sterol biosynthetic pathway (Fig 3C) awaits further investigation. The presence of stigmasta-5,7-dienol and other intermediates (e.g., 4,4-dimethylcholesta-5,7,9-trienol) (Fig 4E) suggests that, possibly, amastigotes accumulated these post-lanosterol intermediates from the host cell sterol pathway and convert them to new sterol products. In five of the HOMEM selected *L. mexicana* AmBR lines and all of the AmBR lines selected in DM, as well as the *L. infantum*, cholesta-5,7,22 trienol became predominant. In cells selected in HOMEM it comprised 86%-93% of the total sterol, while in those cells selected in DM this proportion was somewhat lower (69.7%-75.4%) whilst an isomer, cholesta-5,7,24-trienol ($C_{27}H_{42}O$) increased from 3.4% abundance in wild-type to 7.9%-12.3% in resistant lines selected in DM. In AmBR *L. infantum* 64% of the total comprised this sterol while a further 25% of the total was its isomer cholesta-5,7,24 trienol. Interestingly, cholesta-5,7,22-trienol was higher in wild type cells sustained in DM (16.5%) in comparison with values found in all parental lines cultured in HOMEM (0.5 to 2.57%) (Fig 4A–4C and S5B Table).

## Genomic changes identified with whole genome sequencing in polyene-resistant *Leishmania* spp. lines

Whole genome sequencing was performed on each of the 15 polyene resistant cell lines and particular attention paid to gene deletion or amplification, SNPs and other structural changes found repeatedly in the same gene in different isolates. Given the altered sterol profiles reported in the previous section, we also systematically inspected all genes encoding enzymes of the sterol pathway. All polyene resistant *Leishmania* lines analysed here revealed mutations to one or other of two genes, sterol C5-desaturase (C5DS, gene ID = *LmxM.23.1300*) and sterol C24-methyl transferase (C24SMT, *LmxM.36.2380* and *LmxM.36.2390*), encoding enzymes of the sterol pathway.

### Sterol C5 desaturase mutations

One group of three resistant lines cultured in HOMEM revealed point mutations to C5DS (or lathosterol oxidase (LOX), gene ID = *LmxM.23.1300*). Notably, we identified variants in *Leishmania mexicana* cell lines resistant to both polyenes AmB and for the first time, to the polyene nystatin. In AmBRcl.3, all of the reads of C5DS are mutated (S4 Fig) containing one of two independent heterozygous mutations within the same codon, G220A (V74M) and T221A (V74E), leading to a substitution of a non-polar valine residue, which is conserved across the aligned *Leishmania* species (S5 Fig), by a polar methionine and a negatively charged glutamate residue, respectively. A second homozygous substitution G731T (R244L) was present in AmBRcl.3 and AmBRcl.14 (Fig 5A). Interestingly, both lines also showed a homozygous variant G208A (G70S) that changed a non-polar glycine into a polar serine residue in the adjacent gene downstream from C5DS, a dynein heavy chain (ID = *LmxM.23.1310*) (Fig 5B). Also, in C5DS, AmBRcl.14 showed an in-frame deletion ATG277-279 (M93del) while in NysRcl.B2 a neighbouring frame-shift deletion G286del (A96del) was present (Fig 5A and 5B), leading to loss of a full-length coding region in each case. In all of these C5DS mutants an accumulation of ergosta-7,22-dienol (Fig 4) was observed, indicating that all of these different mutations impacted upon activity of the enzyme leading to an accumulation of its substrate the use of which is ablated in the mutants.

### Sterol C24 methyl transferase mutants

The second sterol pathway gene that was mutated in multiple separate lines was the C24SMT tandem array (*LmxM.36.2380* and *LmxM.36.2390* in *L. mexicana*) (Table 1 and S6 Fig), the mutation of which was previously shown to be common in selection of AmBR. Five AmBR lines of *L. mexicana* cultured in HOMEM, AmBRcl.8, AmBRcl.6, AmBRA4, AmBRB2 and AmBRC1 showed no coverage of the intergenic region (~2.5 kb) between the two copies (S6A and S6B Fig). The *L. infantum* AmB resistant line (AmBRcl.G5) also showed a loss of coverage across this intergenic region (S6E Fig). Copy number variation (CNV) confirmed a deletion within the C24SMT tandem pair in all of these lines (S8 Table). In AmBRA4, a novel non-silent mutation C974T (A325V) resulted in a substitution from alanine to valine. This locus showed no genomic changes in NysRcl.B2 (HOMEM) (S6D Fig). Likewise, in all of the AmB resistant lines selected in DM, this intergenic region was not of decreased coverage (S6C Fig), but a homozygous variant G961A (V321I) was found. Wildtype cells were heterozygous for this variant (Table 1) thus a loss of heterozygosity in lines selected for AmBR in DM appears to have occurred. The accumulation of cholesta 5,7,22-trienol indicates that the duplicated allele is defective, leading to the pathway's truncation at this point in the homozygous state. The mechanism is likely to be analogous to that described previously [96].

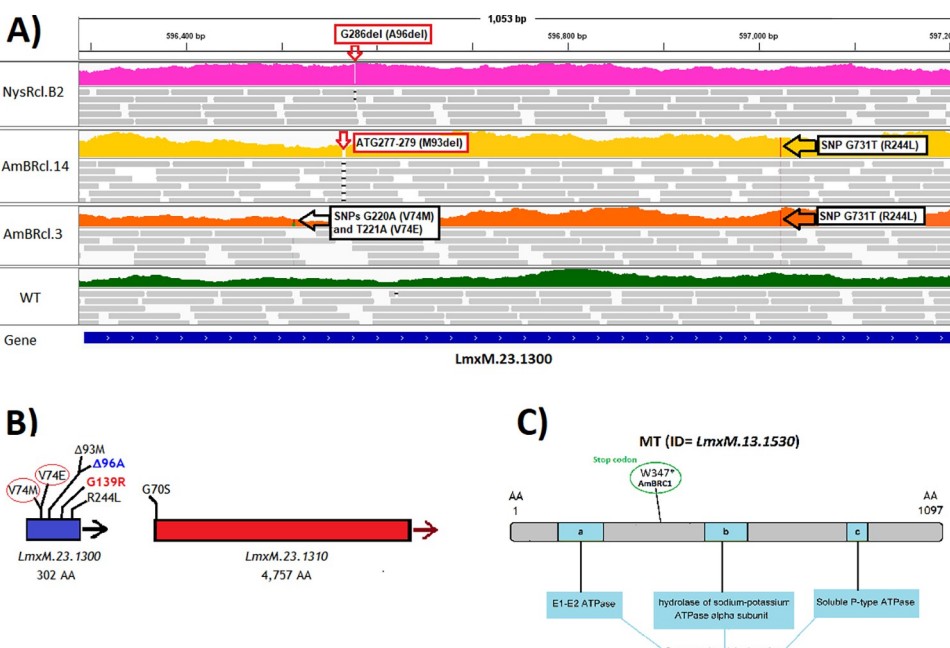

**Fig 5. Mutations in lipid-associated genes found in polyene resistant lines of *L. mexicana*.** Fig 5A) Coverage is shown in colours for NysR (top in pink), AmBRcl.14 (upper-middle in yellow), AmBRcl.3 (lower middle in orange) and wild type (WT) (bottom in green) and reads (grey bars) with mapping quality > 0 are shown for each line. The blue line at the bottom indicates the open reading frame of the C5DS (or LOX) *LmxM.23.1300* gene and variants are indicated (empty arrows). Image produced with WGS data aligned to the reference genome (https://tritrypdb.org) using IGV_2.8.9 (http://software.broadinstitute.org); Fig 5B) Cartoon showing the variants linked to polyene resistance in LOX (C5DS) (blue bar), a SNP in the gene downstream (*LmxM.23.1310*) is also shown (red bar). The red circles indicate that these two mutations are in the same codon and are independent (found in different reads). Mutations found in AmBRcl.14 and AmBRcl.3 (in black) and in Nyscl.B2 (blue) are shown alongside a variant G139R (red) reported in our study [96] which indicates a residue involved in catalysis or ligand binding; Fig 5C) Cartoon of the MT gene *LmxM.13.1530* portraying conserved domains and the proximity of the variant W347* (stop codon) found in this study in AmBRC1 (green circle).

Interestingly, this pre-existent variant was found in AmBR lines (including wild types) cultured in both DM and HOMEM media (Table 1) which can be explained since it constitutes a region that differs between the two copies of C24SMT and is associated with structural changes but not with resistance to AmB [96]. All of the lines where mutations were targeted around the C24SMT locus demonstrated an increase in cholesta-5,7,22-trienol (Fig 4A–4C), which is the substrate for C24SMT, indicating that in each case the mutations have diminished the activity of that enzyme.

## Mutations to the Miltefosine transporter and other genes in multiple resistant lines

Many genes were mutated across the 15 resistant lines, some in multiple derivatives. Several previous studies on resistance to AmB [91,94,96] showed that the miltefosine transporter (MT) gene (*LmxM.13.1530*) was also mutated or deleted during the process of selection of drug resistance. Here we noted that in three lines cultured in HOMEM (AmBRcl.8, AmBRcl.6 and AmBRA4) the MT and an adjacent gene (*LmxM.13.1540*) were deleted (S7C and S7D Fig). In AmBRA4 the deleted region was larger (~20 kb) spanning two additional genes (S7D Fig). Moreover, in one line, AmBRC1, also cultured in HOMEM, a homozygous variant G1041A (W347*) appears (Fig 5C and Table 1), indicating a translation termination (stop

**Table 1. WGS in sterol and non-sterol genes in polyene resistant lines of *Leishmania* spp.** Gene IDs were retrieved from TriTrypDB in *L. mexicana*. Genotype and amino acid changes from variants in sterol genes is provided in S6 Table. A complete list of variants is provided for SNPs (S9 Table) and CNVs (S8 Table). Hom: homozygous; Het: Heterozygous. § This mutation is a pre-existing difference between both copies of the C24SMT gene and is associated with structural changes but not with drug resistance.

| Gene ID and name | Mutation locus | Resistant line(s) with the mutation | Sterol intermediate accumulated | Culture media |
|---|---|---|---|---|
| Sterol genes | | | | |
| *LmxM.23.1300* C-5 desaturase (C5DS) | **1.** G220A (V74M) | AmBRcl.3 (SNPs 1 and 2 are independent) | ergosta-7,22-dienol | HOMEM |
| | **2.** T221A (V74E) | | | |
| | **3.** G731T (R244L) | AmBRcl.3 and AmBRcl.14 | | |
| | **4.** ATG277-279 (M93del) | AmBRcl.14 | | |
| | **5.** G286del (A96del) | NysRcl.B2 | | |
| *LmxM.36.2380 / LmxM.36.2390* C-24 sterol methyl transferase (C24SMT) | **6.** G961A (V321I) § | Hom: AmBRcl.3, AmBRcl.8 and AmBRcl.6; AmBRA4 and LiWT Het: LmWT1, AmBRcl.14, LmWT2, AmBRB2, AmBRC1, LmWT3, NysRcl.B2. | cholesta 5,7,22 | HOMEM |
| | | Hom: AmBRA3-DM, AmBRB2-DM, AmBRC2-DM Het: LmWT-DM | | DM |
| | **7.** Intergenic region with partial or total absence of coverage and decrease in CNV | *L. mexicana*: AmBRcl.8, AmBRcl.6, AmBRA4, AmBRB2 and AmBRC1 (S6A and S6B Fig). *L. infantum*: AmBRcl.G5, LiWT (S6E Fig) | | HOMEM |
| | **8.** C974T (A325V) | AmBRA4 (in *LmxM.36.2390*) | | |
| Non-sterol genes | | | | |
| *LmxM.13.1530* Miltefosine transporter and adjacent genes *LmxM.13.1540* (upstream), and *LmxM.13.1510*, *LmxM.13.1520* (downstream) | **9.** Deletion of a ~9 to 26 kb region spanning various open reading frames (ORF) and a decreased in CNV | AmBRcl.8, AmBRcl.6 and AmBRA4 (S7 Fig). | cholesta 5,7,22 | HOMEM |
| | **10.** G1041A (W347*) | Hom: nonsense mutation (stop gained) in AmBRC1 | | |
| *LmxM.25.2380* Glutathionyl-spermidine synthase (GSS) (LINF_250031200 in *L. infantum*) | **11.** G575C (S192T) | Het: variant in AmBRcl.8 and AmBRcl.6 (*L. mexicana*) and AmBRcl.G5 (*L. infantum*) (S9 Fig) | cholesta 5,7,22 | HOMEM |

codon) and consequent loss of function of MT that resulted in 3.4-fold resistance to MF relative to WT (Fig 1B and S3 Table). Further changes in ploidy were noted in four HOMEM lines. A reduction in CNV in the MT locus was also found in AmBRcl.14, AmBRcl.3 and NysRcl.B2 (they lost one copy of the two) but increased in AmBRB2. CNV analysis confirmed that the MT gene was, however, retained in all of the other lines selected in HOMEM. Notably, in all the lines selected in DM, the MT gene remains unaltered (S7E Fig). Changes in ploidy in all lines are shown in S8 Fig and S8 Table.

Other genes beyond those encoding members of the sterol pathway were also found in two or more polyene-resistant lines and these are listed in Table 1, S10 Fig and S9 Table and their roles or actual relationship with resistance awaits further analysis, although it is notable that four genes encoding enzymes associated with the leishmania thiol redox system (S9B Fig) were found altered. Notably, none of these mutations are present in cell lines selected in DM but they appeared only in HOMEM-mutants in which oxidative stress was more profound (Fig 2). A further gene was found mutated in another HOMEM line, AmBRcl.G5 of *L. infantum*, this novel variant a heterozygous non-synonymous mutation G575C (S192T) (S9A Fig) was found in the glutathionylspermidine synthase (GSS) encoding gene (ID = *LINF_250031200*, formerly annotated as *LinJ25.2500*). Strikingly, the same residue G575C (S192T) in GSS (*LmxM.25.2380*) was also mutated in another two HOMEM grown *L. mexicana* lines,

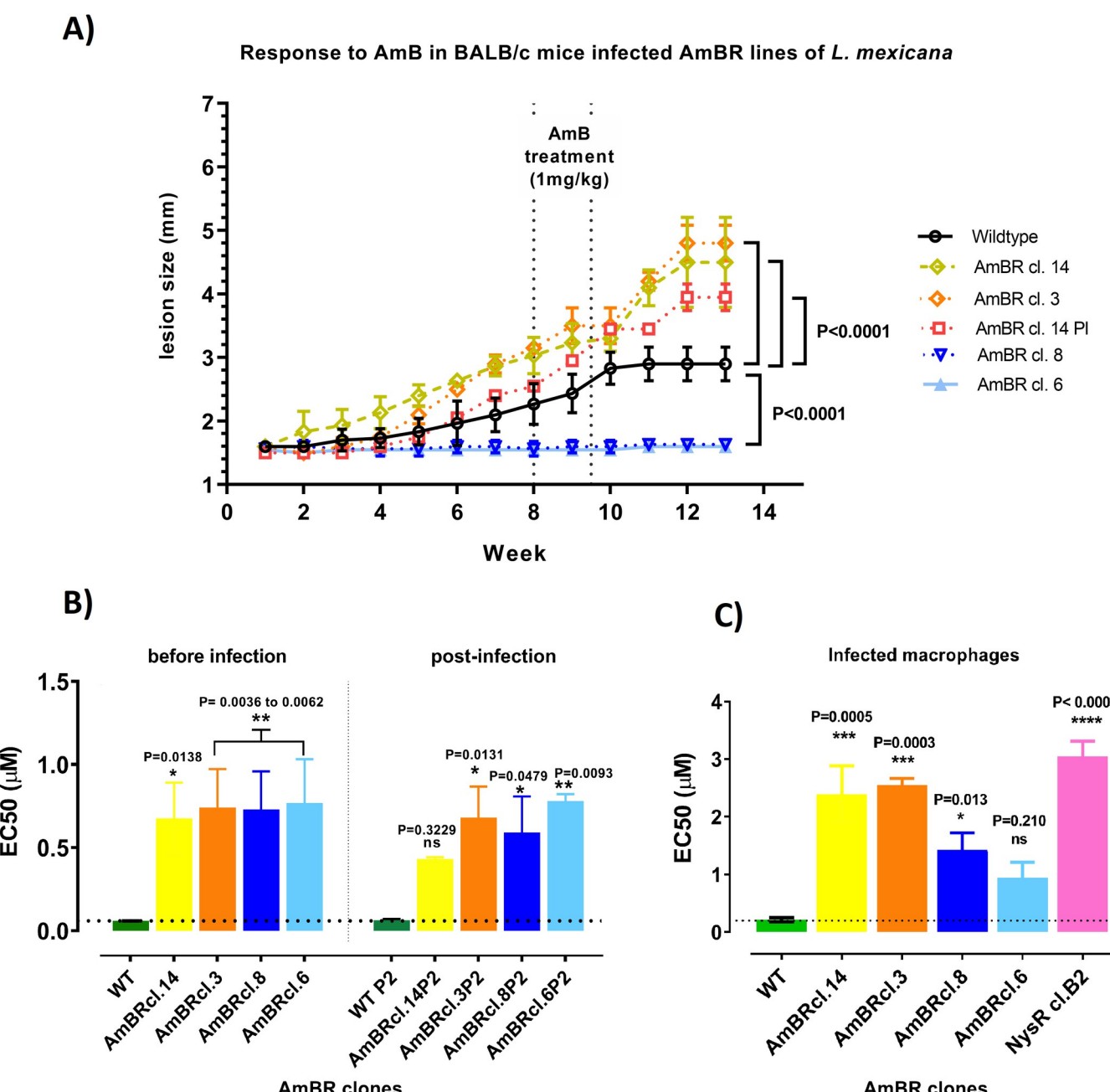

**Fig 6. AmB activity in polyene resistant *L. mexicana*.** Fig 6A) Mice were inoculated in the footpad with stationary phase promastigotes. Evolution of lesion was measured weekly. BALB/c female mice were two-months old at the time of inoculation. Treatment: 1 mg per kg IV every other day in the tail vein with a total of six injections (~120 μl) of AmB (deoxycholate). Fig 6B) activity of AmB in axenic promastigotes before and after infection of mice; Fig 6C) AmB activity against AmBR- and NysR-*L. mexicana* in infected macrophages. Values (μM) are the mean EC50 ± SD (bars) (See S7 Table). Tukey's multiple comparison test was used to find pairwise differences between resistant lines and parental wild type. Statistically significant values (P<0.05, 95% confidence interval) are indicated with stars as follows: P>0.05; * P≤0.05; ** P≤0.01; *** P≤0.001; **** P≤0.0001).

AmBRcl.8 and AmBRcl.6 (S9C Fig) both which showed an attenuated phenotype in mice (Fig 6A) as discussed in the following section. Moreover, in five HOMEM grown lines (AmBRcl.14 and AmBRcl.3, AmBRcl.8, AmBRcl.6 and NysRcl.B2), CNV revealed moderate changes (i.e. they lost one copy) in four genes involved in oxidative stress (S8 Table), two copies of the type II tryparedoxin peroxidase (glutathione peroxidase-like) (*LmxM.26.0800* and *LmxM.26.0810*),

glutaredoxin (*LmxM.26.1540*) and glutathione S-transferase (*LmxM.12.0360*). Previous work has pointed that GSS catalyses the synthesis and degradation of glutathionylspermidine, which is involved in maintaining intracellular thiol redox and in defense against oxidants and the potential role of the leishmanial redox system (S9B Fig) in resistance to AmB [59,82,86–88,90,116,130,131]. The actual role of these genes awaits further investigation to determine a mechanistic explanation and, as noted earlier, evidence on relative sensitivity indicates that polyene resistant lines selected for resistance to AmB in HOMEM, are notably more sensitive to various agents associated with oxidative stress (S3 Table), a feature also noted for AmBR *Candida* strains [76].

## Infectivity of AmBR lines to mice

We initially selected AmB resistance in cultured promastigotes given the tractability of this system for biochemical analysis. However, it is important to understand whether findings from promastigotes translate into amastigote forms that replicate in mammalian macrophages and which are associated with disease. Therefore, we chose four of the HOMEM AmBR mutants to test their ability to retain infectivity and virulence in mice. Two of these were from the C5DS mutant group (AmBRcl.14 and AmBRcl.3) and two from the C24SMT mutant group (AmBRcl.8 and AmBRcl.6). In these C5DS and C24SMT groups, the wild type sterol (ergosta-5,7,24-trienol) was replaced by ergosta-7,22-dienol and cholesta-5,7,22-trienol, respectively. For these experiments, we also included a low passage (P2) isolate (AmBRcl.14 PI) recovered from a previous infection with AmBRcl.14. Notably, mice infected with AmBRcl.14 (C5DS), AmBRcl.3 (C5DS) and AmBRcl.14 PI (C5DS) showed an increase in the size of the lesion at the inoculation site (footpad) up to two times greater (week thirteen) than that observed in mice infected with the parental wild type (Fig 6A). By contrast, mice infected with AmBRcl.8 (C24SMT) and AmBRcl.6 (C24SMT) showed no macroscopic alterations along the entire experiment and no parasites from these two C24SMT mutants were identified using histology, suggesting a fitness loss in these two lines (S11 Fig). Material excised from footpad injection sites and associated lymph nodes from all animals was placed into HOMEM and all yielded proliferative promastigote cultures irrespective of disease progression status in mice. The AmBRcl.14, AmBRcl.14 PI, AmBRcl.3 and wild type lines reached a density of ~1x10$^5$ cells per mL within 72 hours while AmBRcl.8 and AmBRcl.6 required seven days to reach this density. While the lower parasite density obtained with AmBRcl.8 and AmBRcl.6 may be explained by the absence of lesion growth observed in infected mice, the recovery of viable parasites (which also retained their resistance phenotype) from mouse tissue with these two C24SMT mutants was unexpected. Previous work with *L. infantum* showed infectious parasites with a unique sterol profiling [126,132,133]. Here, it appears that accumulation of ergostanes (96.0 to 96.7%) due to loss of sterol C5 desaturase (C5DS or LOX) is related to the hyper-infectivity observed in AmBRcl.14 and AmBRcl.3. Conversely the accumulation of cholestane type sterols through loss of C24SMT (73.3 to 87.9%) in AmBRcl.8 and AmBcl.6 (Fig 4 and S5B Table) may also explain the attenuated phenotype observed in these two mutants. However, it is possible that mutations other than those affecting sterol metabolism can impact on virulence. For example, a missense heterozygous mutation, G575C (S192T), was present in the glutathionylspermidine synthase (GSS) gene of both attenuated lines AmBRcl.8 and AmBRcl.6 (S9C Fig) and in AmBRcl.G5 of *L. infantum* (S9A Fig), but not in AmBRcl.14 and AmBRcl.3 (S10 Fig). Roles for GSS in virulence [134] in *L. mexicana* require further study. It is noteworthy that in *Candida* selected for AmBR *in vitro* changes to sterol composition were associated with resistance, but the phenotype was also associated with enhanced susceptibility to oxidative stress and universal loss of virulence in that study [76].

### Retention of resistance *in vivo* and post infection *in vitro* and in macrophages

We also tested whether the parasites from the four HOMEM lines with mutations in sterol genes and chosen for *in vivo* infection retained resistance to AmB within the mouse (Fig 6A) and when recovered from primary lesions (footpad) (Fig 6B and S7 Table) and lymph nodes. After infection, primary lesion size was monitored weekly thereafter and mice were treated with AmB (six doses of 1 mg per kg given every other day, with higher doses being toxic to mice) at week eight of infection. In wild type parasites, a plateau in lesion size was attained by fourteen days post-treatment and sustained until the end of the experiment (thirteen weeks). By contrast, in mice infected with AmBRcl.14, AmBRcl.3 and AmBRcl.14 PI (the latter recovered from a previous infection) lesion size continued to increase following AmB treatment indicating that these resistant lines were unaffected by this dose of AmB. Due to the absence of increase of lesion in mice infected with AmBRcl.8 and AmBRcl.6 (Fig 6A), retention of resistance in these two lines was assessed *in vitro* as outlined below.

Promastigote cultures of the four HOMEM AmBR lines recovered from footpads and lymph nodes of infected mice treated with AmB were established and $EC_{50}$ values to AmB measured (S7 Table and S12 Fig). Before infection, AmBR isolates were respectively 11.2-fold (AmBRcl.14, P = 0.0138), 12.3-fold (AmBRcl.3, P = 0.0053), 12.1-fold (AmBRcl.8, P = 0.0062) and 12.8-fold (AmBRcl.6, P = 0.0036) less sensitive to AmB than wild type. Comparable resistance was observed in promastigotes recovered post-infection from all AmBR lines and subcultured for two passages (P2) *in vitro* with values of 6.84-fold (AmBRcl.14P2, P = 0.3229), 10.8-fold (AmBRcl.3P2, P = 0.0131), 9.3-fold (AmBRcl.8P2, P = 0.0479) and 12.3-fold (AmBRcl.6P2, P = 0.0093) (Fig 6B). We also tested these four lines that had been recovered from mice (plus the Nys resistant NysRcl.B2 from axenic culture) within infected macrophages and confirmed their resistance phenotype. Interestingly, all lines with mutations in C5DS (and accumulation of ergosta-7,22-dienol) were notably less susceptible (11.20-fold in AmBRcl.14P2, P = 0.0005; 12.2-fold in AmBRcl.3P2, P = 0.0003 and 13.84-fold in Nyscl.B2, P = 0.0001) with respect to their parental wild type (Fig 6C and S7 Table), than either of the two attenuated lines (6.52-fold in AmBRcl.8P2, P = 0.0130 and 4.37-fold in AmBRcl.6P2, P = 0.2100) which had mutations in the C24SMT gene locus (plus a deletion of the miltefosine transporter) in which the accumulation of cholesta-5,7,22 was predominant (Fig 4A). Overall, these findings indicate that the retention of the resistant phenotype was stable across all polyene-resistant lines after being recovered from mice (footpads and lymph nodes) (S12 Fig) irrespective of their genomic variants in sterol genes or their virulence.

## Discussion

AmB has emerged as a leading drug to treat leishmaniasis, particularly in its liposomal formulation AmBisome [8,25,26]. Resistance to the drug would impact heavily on efforts to eliminate VL [40,41]. For leishmaniasis, several instances of AmB treatment failure have been reported in the field, involving cases not known to involve immunosuppression by HIV [59,67] as well as those in HIV infected immunosuppressed patients [47,65,66,135,136]. Moreover, numerous studies have now revealed that resistance can be selected in the laboratory, where the resistance phenotype has frequently been linked to loss of AmB-binding ergostane-type sterols, associated with mutations in enzymes of the sterol pathway. Mutations in the C24SMT gene locus have been common [59,97,99,137], but in other cases C5DS has been implicated [96] and in one case a loss of C14DM (CYP51) was identified [95]. In each case, the mutations lead to loss of ergosterol (ergosta-5,7,24-trienol) and an accumulation of other sterol precursors within the pathway, leading to the conclusion that changes in membrane sterols prevent AmB

binding to the parasite membrane with the same avidity as to WT cells thus leading to loss of sensitivity. Such mutations have been discovered in laboratory-generated AmB-resistant parasites, as well as clinical isolates, of *L. mexicana* [95,96], *L. donovani* [59,97–99] and *L. infantum* [91] which were cultured in media supplemented with 10 to 20% foetal calf serum. The fact that *L. donovani* [59,97–99] and *L. infantum* (reported here) lines that have been selected for resistance to AmB share with *L. mexicana* mutations in sterol pathway enzymes supports the use of *L. mexicana* as a suitable model to discover mechanisms of resistance that are applicable to other *Leishmania* species.

Apart from our previous work in which we analysed four AmBR lines [96], previous studies with AmB have focused on a few mutants which has limited the ability to identify other causes of resistance. Moreover, no previous worked has applied this method in serum and lipid free culture conditions. For this reason, here, we selected a total of 15 new mutants resistant to polyene antimicrobials including ten *L. mexicana* lines resistant to AmB (AmBR) (seven cultured in HOMEM medium containing FBS and three in a serum and lipid free DM) in the hope of expanding knowledge on mechanisms of resistance to AmB. A further four *L. mexicana* lines were selected in HOMEM for resistance to the polyene Nys (NysR), a relative of AmB and a single line of the VL causing, *L. infantum*, was selected for AmBR in HOMEM. Analysis of their respective sterol complements indicated two different profiles had been produced. In both cases, the primary membrane sterol in *Leishmania* spp., ergosta-5,7,24-trienol, was lost and replaced by either ergosta-7, 22-dienol in two of the AmBR *L. mexicana* lines (and in all four NysR lines) selected in HOMEM or by cholesta-5,7,22-trienol in five of the HOMEM selected *L. mexicana* AmBR lines and all three of the AmBR lines selected in DM (AmBR-DM), as well as the *L. infantum* AmBR line (HOMEM). In the former group, mutations to C5DS were identified in sequencing the genome while in the latter group mutations to the C24SMT locus were associated. These observations are consistent with previous studies where C24SMT [59,96–99] and C5DS [96] have been found associated with development of AmBR. Moreover, experiments in *L. major* where C24SMT [103] and C5DS [104] genes were knocked out of WT parasites also showed that viable parasites with resistance to AmB could be recovered. In these two knockouts, however, the effects on other parts of the genome were not determined. In spite of the fact that our findings on fundamental changes to sterol metabolism genes have not found mutations in genes other that C24SMT and C5DS, a number of noteworthy novel findings have been made. For example, we demonstrate that AmB appears to induce oxidative stress in *Leishmania*, as inferred from the large increase of flux through the pentose phosphate pathway. This effect, however, is confined to parasites grown in rich medium with serum added and not in serum free defined medium. In addition, parasites selected for resistance in rich medium including serum reveal multiple mutations in addition to those in the sterol pathway genes, while in DM these additional mutations are not evident.

Since all of this laboratory-based work has focused on promastigote forms of the parasites, which are amenable to simple manipulation, questions as to their relevance *in vivo* have been made. Purkait et al. [59] isolated parasites from a VL patient who failed treatment and accumulation of sterols indicated changes to C24SMT similar to those found by others in mutants generated under laboratory conditions [96–99]. This indicates the potential of C24SMT mutants to propagate as AmBR cells in patients. *L. major* C24SMT-KO mutants derived by Mukherjee et al. [103] were of reduced virulence in mice, as were the two *L. mexicana* lines AmBRcl.8 and AmBRcl.6 with C24SMT mutations we tested here. However, Pountain et al. [96] earlier found a C24SMT mutant that retained virulence in mice. This indicates that it might be mutations other than those to C24SMT *per se* that are responsible for attenuation in mice. Here, for example, we identify changes to the glutathionylspermidine synthase (GSS) gene in both attenuated *L. mexicana* lines (AmBRcl.8 and AmBRcl.6) and in *L. infantum*

(AmBRcl.G5) (S9A Fig), although whether these changes are associated with attenuation is not yet known and requires further analyses. While some lines had lost virulence, two separate lines selected here (AmBRcl.14 and AmBRcl.3) with variants in C5DS, both retained virulence and resistance to AmB in mice. In *L. major* C5DS-KO lines displayed only a small loss of virulence in mice [96,104], thus C5DS mutations can also be selected, yielding parasites that are both AmBR and still able to infect mammals. Earlier work in *Candida* [76] had noted loss of virulence associated with selection of AmBR as a universal feature. In *Leishmania*, however, the risk of selection of AmBR in *Leishmania* appears to be a genuine risk, albeit the selective pressures on amastigotes in treated patients being distinct from those applied to selecting resistance in cultured promastigotes. Since AmB resistance mechanisms often appear to be similar between *L. donovani* and *L. mexicana* [59,82,87–89] the risk of AmB resistance in visceral leishmaniasis must also be considered as real.

Most studies in AmBR fungi, particularly yeasts including *S. cerevisiae* and *Candida* spp. also showed a tendency for C24SMT and C5DS (ERG6 and ERG3 in fungi, respectively) [74,76] mutations although changes in other genes of the sterol pathway (ERG- in fungi) such as C14DM (ERG11) [72,76,138,139], C22-sterol desaturase (ERG5) [72,74] and C-8 sterol isomerase (ERG2) [71,76] have also been associated with resistance. In yeast, contradictory findings have been made regarding the role of oxidative stress. For example, enhanced oxidative stress resistance has also been implicated in loss of susceptibility to AmB in some studies [45,83,84], while increased sensitivity to oxidative stress was seen as a universal feature in *Candida* strains selected for AmBR [76]. In *Leishmania*, other studies also pointed to oxidative stress response in AmBR [59,82,86,88,90,116,130]. Here we show that the activity of AmB appears to be substantially augmented in serum and lipid rich medium (FCS supplemented HOMEM) and associated with enhanced induction of oxidative stress (including exposure at high concentrations for 15 mins) as judged by increased flux of glucose via the PPP in AmB treated cells. The activity of AmB against wild type *L. mexicana* grown in HOMEM containing FBS compared with serum-free DM showed the drug to be significantly more potent (3.6 to 4.11-fold) in the rich conditions (Fig 3A, 3B and 3D and S3 Table) which may point to enhanced oxidative stress causing this increased influx to regenerate NADPH lost in fuelling the redox defence pathways [116–118]. Notably, too, beyond the loss of heterozygosity associated with C24SMT, parasites selected for AmB resistance in DM did not reveal additional genomic changes in any redox pathway genes as found in five AmBR lines (AmBRcl.14 and AmBRcl.3, AmBRcl.8, AmBRcl.6) and the nystatin resistant (NysRcl.B2), sustained in HOMEM, emphasising that environment-specific impacts play key roles in AmB mode of action and resistance. However, our data indicated that AmBR is associated with increases in sensitivity to oxidative stress, as previously reported for *Candida* [76] where possible compensatory changes enabling optimal growth after altering sterol metabolism may have occurred. Further probing of the combined changes to the system are needed to understand the relationship between parasites, sterol metabolism, oxidative stress and responses to AmB.

In all the lines selected here in HOMEM, as well as many earlier selected lines (also cultured in the presence of serum), genes encoding enzymes involved in sterol metabolism are mutated. Three of the 15 lines selected here had also deleted the miltefosine transporter (MT), a lipid flippase. Some earlier studies had also reported this loss [94,96] and a study seeking genes whose down regulation could underpin AmBR in *T. brucei* also found a role for the orthologous gene in this species [140]. It is possible the transporter plays a direct role in AmB acquisition, or else it is involved indirectly through changes in membrane architecture which are not present in serum free conditions that impact the interaction of AmB with the membrane. In our previous work we observed higher levels of resistance to AmB achieved through the deletion of the MT followed by structural rearrangement in the C24SMT locus [96]. Here, we did

not study the appearance of mutations across a time course, but it is possible that selection in low level AmB leads to changes in genes such as the MT, for example, in some lines before the acquisition of higher-level resistance through loss of drug binding sterols derived from mutations to genes involved in sterol synthesis. Earlier appearing mutations in the latter would preclude a need for mutations in the flippase since cells would have already exceeded the resistance level available through flippase-loss.

While this study is focused on changes to sterol metabolism, and a loss of virulence was noted for C24SMT mutants (AmBRcl.8 and AmBRcl.6), we cannot rule out that the loss of virulence is not directly associated with the sterol changes. For example, additional mutation in GSS and a deletion of the MT was noted in each. Further study is required to understand the relative contributions of each. However, such mutations are found in both *L. mexicana* and *L. infantum* so it is possible such changes might be required to enable changes to sterol metabolism when grown in rich medium as reported for *Candida* [76]. Unlike the *Candida* study, however, leishmania selected in this way can retain virulence in mice. Two C5DS mutant lines retained their infectivity in mice. In these, two different independent mutations in the same residue (V74M) and two deletions (M93del and A96del) alongside the accumulation of ergosta-5,7,24-trienol, an ergostane-type sterol is associated with AmB resistance and virulence [55,96,126], were found. Previously a C24SMT associated mutant also retained virulence [96].

The fact that AmBR has been rare in the field until now might simply be due to the fact that, until recently, it was not widely used, and has generally involved protracted dosing regimens. Extended use and the decision to use a single dose of the drug in monotherapy has great public health benefits [40,41,141,142] but that policy must consider the fact that selection of AmB resistance is clearly feasible in both CL and VL species, as we have shown here with *L. mexicana* (CL) and *L. infantum* (VL), and that resistance can be sustained in parasites that are virulent in mammals. Although CL is mainly treated with antileishmanials other than AmB, the Pan American Health Organisation has suggested its use in relapse cases for New World cutaneous species [143] although several reports indicate that liposomal AmB was unsatisfactory in treating CL caused by *L. mexicana* [144–146].

Another important point for consideration relates to the widespread administration of miltefosine in clinical populations which may carry a risk of cross resistance between AmB and MF [91–94] given that it seems loss of the miltefosine transporter can contribute to AmBR. Interestingly, here, as in other studies, AmB resistance is accompanied by increased sensitivity to the antileishmanials pentamidine and paromomycin. Data from *L. major* have suggested changes in mitochondrial membrane potential [102] related to lipid remodelling [111] might underlie pentamidine hypersensitivity, but mechanisms that could explain paromomycin hypersensitivity are not yet known. The possible use of pentamidine and paromomycin against AmBR lines should they become problematic in the field, however, deserves further consideration.

## Material and methods

### Ethics statement

Mice (from Harlan UK Ltd) were maintained at the Central Research Facilities of the University of Glasgow, Glasgow U.K and euthanized following the United Kingdom regulations and the Animals (Scientific Procedures) Act, 1986 (ASPA) and animal protocols and procedures were approved by The Home Office of the UK government and the University of Glasgow Ethics Committee under licence PCF371688.

## Cell culture and polyene-resistant selection

*L. mexicana* (MNYC/BZ/62/M379) and *L. infantum* JPCM5 (MCAN/ES/98/LLM-877) pro-mastigotes were cultured at 25°C and 27°C, respectively, in HOMEM (GE Healthcare and Gibco) supplemented with 10% v/v heat-inactivated foetal bovine serum (HI-FBS) (Labtech International) or in serum-free defined medium (DM) [147,148]. HOMEM and DM culture media (S2 Table) were supplemented with 100 μg per mL streptomycin, and 100 IU mL−1 penicillin. Parasites were sub-cultured weekly starting at a density of 1 x $10^5$ cells per ml and growth rate, cell density and morphology were determined by light microscopy with a haemo-cytometer [149,150]. S1 Table shows all the fifteen-individual polyene-resistant lines selected *in vitro* alongside their respective parental wild type, cultured in parallel in the absence of drug. *L. mexicana* cells were selected for resistance by stepwise increasing concentrations of AmB (ten lines) in either HOMEM (thirty-two weeks) starting from 50 nM and rising to 220–300 nM (S1A and S1C Fig) or in serum-free defined medium DM (fifty weeks) from 100 nM up to 300 nM (S1D Fig). Likewise, AmB (one line) for *L. infantum* in HOMEM (fifteen weeks) started at 50 nM up to 100 nM (S1E Fig). Nystatin was added to *L. mexicana* (four lines) in HOMEM from 1.5 μM—up to 12.5 μM (S1C Fig). Individual clones from each independent resistant line were obtained by limiting dilution assay (LDA). The most resistant clones (n = 15) (S1 Table) were selected for further analyses. Retention of resistance (as promasti-gotes) of all independent lines was confirmed after at least ten passages in their respective culture media (HOMEM or DM) without drug pressure and post-infection *in vivo* in all resistant lines (n = 5) and wild type included in these experiments. Cell lines were preserved in culture medium with 15% DMSO (v/v). Axenic amastigotes were cultured at 32.5°C with 5% $CO_2$ in Schneider's medium (SDM) pH 5.5, supplemented with HI-FBS (10% v/v), 1.5 mL of hemin (2.5 mg per mL in 50 mM NaOH, 0.003% v/v), 100 μg per mL streptomycin, and 100 IU mL$^{-1}$ penicillin as described before [151].

## Drug susceptibility assays

Drug susceptibility to various drugs was determined using the Alamar Blue assay [152–154] with some modifications. Briefly, mid-log phase promastigotes (1 x $10^6$ cells per mL) were incubated for 72 hours with different concentrations of drugs serially diluted in a two-fold stepwise fashion in 96-well plates, followed by the addition of resazurin (0.49 mM in 1x phos-phate-buffered saline (PBS), pH 7.4) and further incubation for 48 hours. For the experiments assessing retention of resistance within macrophages, incubation with AmB (AmBisome) and resazurin was for 48 and 6 hours, respectively. Absorbance (fluorescence) was measured using a BMG LabTech Fluostar Optima fluorometer. Intensity was read at $\lambda_{EX}$ 530 nm and $\lambda_{EM}$ 590 nm and analysed with Prism 8.0 software to obtain the 50% inhibitory concentration ($EC_{50} \pm$ SD, 95% confidence interval) using regression analysis. One-way ANOVA and Tukey's multi-ple pairwise comparison tests were performed independently for each compound. All assays were performed in three biological replicates. All drugs used in these assays were bought from Sigma-Aldrich.

## Infection of mice and macrophages

Eight-week-old female BALB/c mice (from Harlan UK Ltd) were kept at the Central Research Facilities of the University of Glasgow, Glasgow U.K and randomly assigned to either of the treatment groups (wild type versus AmBR lines). Infection of mice was performed by inoculat-ing 2 x $10^6$ stationary phase *L. mexicana* promastigotes resuspended in 100–200 μl of filter ster-ilised PBS in the left footpad. Mice were treated with six doses of intravenously injected AmB (1 mg per kg) into the tail vein q.o.d. (every other day) starting at week 8 of infection as

described elsewhere [120]. Lesion size was assessed weekly before and after chemotherapy (thirteen weeks in total). Amastigotes were recovered post-infection from footpad- and lymph node tissue and sub-cultured in Schneider's medium (SDM) as previously described [151], or transformed into promastigotes and maintained in complete HOMEM. Footpad- and lymph nodes were processed for histology and stained with haematoxylin and eosin (H&E) at the Beatson Institute, University of Glasgow, and analysed with light microscopy.

### Retention of resistance to AmB post-infection in mice

Retention of resistance to AmB after infection in mice (NysRcl.B2 was from axenic promastigotes culture) was assessed in both axenic promastigotes and in infected macrophages. In the first case, a low passage (P2) of axenic parasites was recovered from mouse tissue (as amastigotes) and transformed into promastigotes in HOMEM and drug susceptibility was performed as with assays described above. Retention of AmB resistance within infected macrophages was performed following a method described elsewhere with some modifications [155–157]. Briefly, a macrophage-like cell line (RAW 264.7) initially derived from BALBc/mice was maintained at 37˚C in a humidified atmosphere with 5% $CO_2$ in Dulbecco's Modified Eagle's Medium (DMEM) with 4.5 g/L glucose, +L-glutamine (584 mg/L) and 110 mg/L Sodium Pyruvate (Gibco), supplemented with 10% HI-FBS, along with 100 IU and 100 µg/ml each of penicillin and streptomycin. Stationary phase promastigotes ($2.5 \times 10^6$ per mL) were used to infect adherent macrophage-like cells ($2.5 \times 10^5$ per mL) at a ratio 10:1 for 24 hours. Infected cells were then washed (3 x) with DMEM to remove extracellular parasites before the addition of variable concentrations of AmB and further incubation for 48 hours. DMEM was then removed, and macrophages were lysed using sodium dodecyl sulfate 0.05% (20 µl per well) for 1 min followed by resuspension in complete HOMEM (180 µl per well) and plates were incubated at 26˚C for 48 hours. The presence of promastigotes was assessed with direct microscopy prior to determination of the $EC_{50}$ as described before for drug assays.

### Sterol analysis by GC-MS

GC-MS was performed at Glasgow Polyomics. Sterols were extracted from parasite pellets for Gas chromatography-mass spectrometry (GC-MS) analysis. Mid-log ($3 \times 10^8$) axenic promastigotes and amastigotes (recovered post infection and maintained in SDM as described previously) [151,158] were washed in PBS and resuspended in 500 µl of KOH-ethanol (20 ml $dH_2O$ in 30 ml EtOH, 12.5 g KOH) and incubated at 85˚C for 1 hour in Pyrex glass tubes. A similar volume of n-heptane was added, and samples were vortexed for 30 seconds. After 20 min the organic layer containing the sterols was transferred into glass borosilicate vials with a Teflon cap (Thermo Fisher) and stored at -80˚C until GC-MS analysis. All samples including blanks (no cells) and pooled samples (a mix with 10–20 µl from each sample) were in triplicate. Sterols were detected as ester derivatives with trimethyl silane (TMS) and were reported as their underivatized form after normalisation. Briefly, samples were dried with $N_2$ flow at 60˚C in glass vials followed by addition of 50 µl of N-methyl-n-trimethylsilyltrifluoroacetamide with 1% 2,2,2-trifluoro-N-methyl-N-(trimethylsilyl)-acetamide, chlorotrimethylsilane (Thermo Scientific), vortexed for 10 secs followed by incubation at 80˚C for 15 mins and allowed to cool down at RT. After this, 50 µl of pyridine and 1 µl of the retention index solution were added and samples were vortexed for 10 secs. GC was performed in a TraceGOLD TG-5SILMS column with 30 m length, 0.25 mm inner diameter and 0.25 µm film thickness installed in a Trace Ultra gas chromatograph (both from Thermo Scientific) with helium flow rate of 1.0 ml/min, then 1 µl of TMS-derivatised sample was injected into a split/splitless (SSL) injector at 250˚C using a surged splitless injection -splitless time of 30 secs and a surge pressure of 167 kPa. Temperature was

increased from 70- up to 250˚C at a ramp rate of 50˚C/min followed by a ramp rate reduction of 10˚C/min reaching a final temperature of 330˚C which was maintained for 3.5 min. Eluting peaks were transferred at an auxiliary transfer temperature of 250˚C to a ITQ900-GC mass spectrometer (Thermo Scientific), with a filament delay of 5 min. Electron ionisation (70 V) was used with an emission current of 50 μA and an ion source that was held at 230˚C. The full scan mass range was 50–700 m/z with an automatic gain control (AGC) of 50%, and maximum ion time of 50 ms. Samples, blanks, pools and standards were loaded into the instrument. Analysis was performed with a TraceFinder v3.3 (Thermo Scientific) and sterol peaks were identified by matching the standards or by comparison to the NIST library.

## Untargeted metabolomics by LC-MS

Mid-log *L. mexicana* promastigotes (1 x $10^8$) of all AmB-resistant lines alongside their parental wild-type were treated with AmB (Sigma) at 5 x their $EC_{50}$ for 15 mins as follows: the amount of AmB for lines selected in HOMEM was 0.3 μM for both parental lines (LmWT1 and LmWT2), 3 μM for AmBRcl.14 and AmBRcl.8, and 1.74 μM for AmBRB2. Both wild type (LmWT-DM) and AmBRA3-DM cultured in DM media were treated with AmB at 1.16 μM and 3.5 μM, respectively.

Metabolite extraction was performed by quenching parasite pellets by rapid cooling at 4–10˚C in dry ice/ethanol bath, culture medium was removed by centrifugation at 1,250 g for 10 minutes at 4˚C, then transferred and centrifuged twice at 4,500 rpm for 10 minutes at 4˚C with a wash in 1ml of cold PBS in between and the pellet was resuspended in 200 ul of monophasic chloroform/methanol/water (CMW 1:3:1) followed by 1 hour shaking (15,000 rpm) at 4˚C and a final spin at 13,000 rpm for 10 minutes at 4˚C. Pooled sample was made by taking 10 μl from each sample and blanks were also included. Samples were sealed after adding argon and stored at -80˚C until analysis with LC-MS. Separation of metabolites by LC-MS was performed with a ZIC pHILIC column (150 mm × 4.6 mm, 5 μm column, Merck Sequant) coupled to high-resolution Thermo Orbitrap QExactive (Thermo Fisher Scientific) mass spectrometry in both positive and negative ionization modes. Samples were analysed in four replicates and data were processed at Glasgow Polyomics and provided as raw data. Identification of Liquid Chromatography-Mass Spectrometry (LC-MS) peaks involved mzmatch [159] run though PiMP [115] and IDEOM [160–162] and analysis was conducted in these latter software packages. LC-MS raw files were submitted to the repository MetaboLights [163] (https://www.ebi.ac.uk/metabolights/) with project number MTBLS2744.

## Genomic DNA extraction and sequencing

Genomic DNA was extracted with the Nucleospin Tissue kit (Macherey-Nagel) from polyene-resistant and wild-type mid-log promastigotes (1 x $10^7$) washed twice in PBS at 1,250 g for 10 minutes. Library prep and sequencing were performed at Glasgow Polyomics using an Illumina NextSeq 500 sequencer yielding 2 x 75 bp paired-end reads. Reference genomes of *L. mexicana* MHOM/GT/2001/U1103 and *L. infantum* JPCM5 (MCAN/ES/98/LLM-877) release 46 were obtained from the TriTrypDB (https://tritrypdb.org). Reads were mapped to their reference genome using BWA-MEM [164] and PCR duplicates were removed using GATK version 4.1.4.1 [165]. Variant calling was performed using MuTect2 [166] with the default settings and variant annotation using snpEff [167]. Filtered Variant Call Format (VCF) files corresponding to SNPs and indels with an impact on coding sequences were compared between wild type and resistant lines and manually verified using the IGV 2.8.9 visualization tool [168] and eliminated if present in the parental line. Copy ratio alterations were detected using GATK (version 4.2.0.0). Reference genomes were divided into equally sized bins of 1,000 base

pairs using the PreprocessIntervals tool. Read counts in each bin were collected from alignment data using the CollectReadCounts tool. Copy ratios of a resistant sample over a matched non-resistant sample were obtained from read counts using CreateReadCountPanelOfNormals and DenoiseReadCounts tools. Copy number ratio was obtained by comparing the resistant lines with the corresponding copy number of wild type used as the base line. Genome contigs were segmented using ModelSegments tool from copy ratios of resistant and non-resistant samples. Amplified and deleted segments were identified using CallCopyRatioSegments tool with the default settings. Plots of denoised and segmented copy-ratios were generated using R software. Sequencing raw data was deposited at the NCBI National Center for Biotechnology Information (https://www.ncbi.nlm.nih.gov) with project PRJNA763929 and PRJNA770503.

## Other computational tools

BLAST search was performed against the NCBI database using default settings. ClustalW Omega was used to identify and align sequences. Protein queries-sequences were retrieved from TriTrypDB, Uniprot (https://www.uniprot.org/) and the Protein Data Bank (PDB) (https://www.rcsb.org/). Protein alignment viewers used were Clustal X2 [169]. Protein-protein interaction map was created using STRING Database (https://string-db.org) [170–173].

## Supporting information

**S1 Table. List of polyene resistant lines selected for downstream analyses.** A) List of the most resistant clones chosen from each line. B) List of individual clones selected for downstream analyses.
(XLSX)

**S2 Table. Formulation of HOMEM and DM culture media.** A) HOMEM. B) Defined Medium (DM). Medium supplemented with serum and antibiotics as described in Material and Methods.
(XLSX)

**S3 Table. Drugs susceptibility profile (EC$_{50}$ values and Fold-change) to the antileishmanials in all polyene resistant lines of *Leishmania* spp.** Mean EC$_{50}$ values (μM) ± Standard Deviation (SD). Except for NMC tested in duplicates, values are from at least three biological replicates. One-way ANOVA was performed independently for each compound to determine differences of the mean between groups. Tukey's test compared pairwise differences of resistant lines with respect the parental wild type. Statistical difference (P<0.05, 95% Confidence Interval) is shown with stars as follows: ns non-significant or P>0.05; * P≤0.05; ** P≤0.01; *** P≤0.001; **** P≤0.0001.). ND: not determined. AmB (amphotericin B), NMC (natamycin), Nys (nystatin), keto (ketoconazole), MF (miltefosine), PAR (paromomycin), PAT (antimony potassium tartrate), PENT (pentamidine), MB (methylene blue). See Material and Methods for full details. Fold change in EC$_{50}$ is relative to the parental to the respective parental wild type cultured in parallel without drug. Values higher and lower than 1, indicate a decrease and increase in susceptibility, respectively.
(XLSX)

**S4 Table. LC-MS data of AmBR lines cultured in HOMEM and DM.** Metabolites in the pentose-phosphate pathway detected with LC-MS in mid log wild type parasites (1 x 10e8) cultured in HOMEM or DM and treated with high concentrations of AmB for 15 mins (See Material and Methods for a full description). Multivariate data analysis was performed with PiMP analysis pipeline [115] and the Benjamini-Hochberg procedure adjusted raw P-values

(q-values) < 0.05 for ANOVA. Each experiment represents one biological replicate (n = 4). (XLSX)

**S5 Table. Sterol peaks identification and composition (%) found by GC-MS in polyene (AmB and Nys) resistant lines of *Leishmania* spp.** A) Identification of sterol peaks by GC-MS. Standards used were: cholesterol, TMS; desmosterol, TMS; 5α-cholest-7-en-3β-ol, TMS; ergosterol, TMS; stigmasterol, TMS; β-sitosterol, TMS; lanosterol, TMS; FF-MAS (4,4-dimethyl-5α-cholesta-8,14,24-trien-3ß-ol) and zymosterol. FF-MAS: Follicular fluid meiosis-activating sterol and an intermediate in the cholesterol biosynthetic pathway present in all cells. Peaks that did not match any standard were determined by comparing with the NIST spectral libraries with the ion trap mass spectrometer. 5-dehydroepisterol based on literature [100,126,132,174]. RT: retention time. TMS: trimethylsilyl (for derivatization). B) Sterol composition (%) by GC-MS in polyene (AmB and Nys) resistant lines of *Leishmania* spp. promastigotes and amastigotes. Content of each sterol is the percentage of the total of the raw peak area detected per line ± Standard deviation of three independent biological replicates. Sterol content is shown as the total (%) of the raw peak area detected per sample (± SD) from all biological replicates (n = 3).
(XLSX)

**S6 Table. Genotype of mutation identified in sterol genes in polyene resistant and parental wild type of *Leishmania* spp.** LOX (C5DS): Lathosterol oxidase a.k.a. C-5 desaturase (*LmxM.23.1300*); * The genotype in both C24SMT copies is G961/A961 for *LmxM.36.2380* & *LmxM.36.2390* (*LINF_360031200 & LINF_360031300* in *L. infantum*), respectively. ‡ these mutants showed low or no coverage of the intergenic region between both copies of C24SMT (S6 Fig). ND = Non determined due to total absence of coverage at this locus. Changes in the miltefosine transporter (MT) *LmxM.13.1530* are included here and the visualization of genome coverage at this locus is shown in S7 Fig.
(XLSX)

**S7 Table. Susceptibility to AmB of AmBR *L. mexicana* axenic promastigotes before and after infection in mice and in infected macrophages.** Values in µM, Mean ± Standard Deviation (SD). One-way ANOVA was performed independently for each compound to determine differences of the mean between groups. Tukey's multiple compared pairwise differences of resistant lines with respect the parental wild type. Statistical difference (P<0.05, 95% Confidence Interval) is shown with stars as follows: ns non-significant or P>0.05; * P≤0.05; ** P≤0.01; *** P≤0.001; **** P≤0.0001. FC: Fold Change of AmBR lines is relative to their respective parental wild type.
(XLSX)

**S8 Table. Copy Number Variation (CNV) in all polyene resistant lines of *Leishmania* spp.** Copy number ratio was obtained by comparing the resistant lines with the corresponding wild type line which is used as the base line. Gene ID (column A-I) were obtained from TriTrypDB-46. Reference genomes were divided into equally sized bins of 1000 base pairs and the start and end points of the called segments were approximate and not accurate (see Material and Methods for a full description). A visualization of this Table is shown in S8 Fig.
(XLSX)

**S9 Table. Protein altering mutations (SNPs) identified in all polyene resistant lines of *Leishmania* spp.** Paired end reads were aligned with the corresponding wild type parental line. Reference genome and gene IDs were retrieved from TriTrypDB database (see Material

and Methods for a full description). The most relevant mutations are listed in Table 1.
(XLSX)

**S1 Fig. Selection pressure in polyene-resistant lines of *Leishmania* spp. promastigotes.** *L. mexicana* (S1A to S1D Fig) and *L. infanum* (S1E Fig) mid-log promastigotes ($5x10^5$) were growth with a stepwise increasing concentration of AmB (S1A, S1C-E Fig) or Nys (S1B Fig) added in the cultured medium and indicated with coloured pipelines (left y-axis). The $EC_{50}$ (right-y axis) is shown for wild type (black bars) and resistant lines, AmBR and NysR (coloured bars in all panels). The mean $EC_{50}$ of the parental wild type is shown (horizontal dotted lines). See Material and Methods for a detailed description.
(TIF)

**S2 Fig. Growth rate and size of polyene resistant *L. mexicana* promastigotes.** Cell density was measured every 24 hours for 8 days starting from 1x10e5 cells/ml. S2A Fig) AmBR lines in HOMEM. S2B Fig) NysR in HOMEM. S2C Fig) AmBRA4 in HOMEM and AmBRA3-DM in Defined Medium (DM). S2D Fig) Violin plot of the mean cell body length (μm) of four AmBR lines from panel A showing AmBRcl.14 and AmBRcl.3 with an increased length relative to wild type. The central continuous line within each coloured violin plot is the median of each group. The cell body length was measured from the base of the flagellum until the posterior endpoint of the cell body of promastigotes in the stationary phase. Data were processed with ImageJ software and represent the mean of the sample ($n \geq 30$). Measurements are the mean (±SD) of three biological replicates. Tukey's multiple comparison test was used to find pairwise differences between resistant lines and parental wild type. Statistically significant values ($P < 0.05$, 95% Confidence Interval) are indicated with stars as follows: $^*P \leq 0.05$, $^{**}P \leq 0.01$, $^{***}P \leq 0.001$, $^{****}P \leq 0.0001$).
(TIF)

**S3 Fig. Comparison of metabolites between cells grown in DM and HOMEM.** Metabolites in the glycolysis (S3A Fig), citric acid cycle (TCA) (S3B Fig) and Pentose Phosphate Pathway (PPP) (S3C Fig) detected with LC-MS in mid log wild type parasites (1 x 10e8) cultured in HOMEM or DM. Multivariate data analysis was performed with PiMP analysis pipeline [115] and the Benjamini-Hochberg procedure adjusted raw P-values (q-values) < 0.05 for ANOVA. Each experiment represents one biological replicate (n = 4).
(TIF)

**S4 Fig. Visualization of the genomic region in the sterol gene *LmxM.23.1300* in AmBRcl.3.** The image shows two independent heterozygous mutations, G220A (V74M) and T221A (V74E) in AmBRcl.3. The red square indicates the localisation of both mutation in different reads. Image was produced with WGS data aligned to the reference genome (https://tritrypdb.org) using the IGV_2.8.9 software (http://software.broadinstitute.org/software/igv/). See Fig 5 for a full description of the panels.
(TIF)

**S5 Fig. Alignment of the *Leishmania* gene lathosterol oxidase (or sterol C5-desaturase).** *L. mexicana* gene *LmxM.23.1300* (LOX). Black boxes and stars denote His and other residues conserved across species, respectively. Novel variants identified in this study are indicated with black (AmBR lines) and red (NysR line) arrows. Also included (blue box) is the variant from our previous study [96]. From top to bottom kinetoplastids listed are *L. mexicana*, *L. major*, *L. infantum*, *L. braziliensis*, *T. brucei*, *T. cruzi*, *Bodo saltans* and *Crithidia fasciculata*. Also included are *D. rerio* (zebra fish), *M. musculus* (mouse), *Homo sapiens* (human), *S. cerevisiae* (budding yeast), *C. albicans* (pathogenic fungi) and *Arabidopsis thaliana* (plant). Proteins

sequences are from TriTrypDB (https://tritrypdb.org) and Uniprot (https://www.uniprot.org/). Alignment was performed using Clustal Ω with default settings (https://www.ebi.ac.uk/Tools/msa/clustalo/).
(TIF)

**S6 Fig. Coverage of *LmxM.36.2380* and *LmxM.36.2390* in polyene resistant lines cultured in HOMEM and DM.** S6A-B Fig) AmBR lines (and their respective parental WT) cultured in HOMEM showing the absence of coverage in this locus in five lines. S6C Fig) AmBR lines and wild type cultured in DM. S6D Fig) Reads coverage in the intergenic region in NysRcl.B2 and its parental WT. S6E Fig) *L. infantum* showing total absence (AmBRcl.G5) and partial coverage (parental WT). Reads were aligned to the reference genome (https://tritrypdb.org) and the image was produced using the software IGV 2.8.9 (http://software.broadinstitute.org/software/igv/). See Fig 5 for a full description of the panels.
(TIF)

**S7 Fig. Coverage of the miltefosine transporter of (MT) and adjacent genes in *Leishmania* spp.** The coordinates of a region spanning a genomic region of ~16 kb and ~26 kb are shown. Genes are shown at the bottom (blue bars). S7A-B Fig) coverage in NysRcl.B2 and AmBRcl.G5 S7C Fig) In AmBRcl.8 and AmBRcl.6 a total absence of coverage of a region (~9 kb) comprising *LmxM.13.1530* and adjacent gene downstream *LmxM.13.1540* is shown while small gaps of low coverage are observed downstream from each gene in AmBcl.14, AmBcl.3 and WT. S7D Fig) Absence of a ~20 kb region comprising four genes (from *LmxM.13.1510* to *LmxM.13.1540*) is shown in AmBRA4. WGS data were aligned to the reference genome (https://tritrypdb.org) and the images were produced with IGV_2.8.9 software (http://software.broadinstitute.org/software/igv/). See Fig 5 for a full description of the panels.
(TIF)

**S8 Fig. Copy Number Variation (CNV) in all polyene resistant lines of *Leishmania* spp.** Copy ratio alterations were detected using GATK (version 4.2.0.0). Reference genomes from TriTrypDB46 (https://tritrypdb.org) were divided into equally sized bins of 1000 base pairs (start and end points of the called segments were approximate). CN ratio is obtained by comparing the resistant lines with the corresponding wild type line. Plots of denoised and segmented copy-ratios were generated using R. The thick black lines show the mean copy ratios of corresponding segments. Values of this Fig are provided in S8 Table.
(TIF)

**S9 Fig. Visualization of the variants in glutathionylspermidine synthase (GSS).** S9A Fig) Genomic region showing a heterozygous mutation (boxed) in GSS in AmBRcl.G5 of *L. infantum*. S9B Fig) Protein-protein interaction (PPIs) map of GSS and other enzymes relevant for the trypanothione biosynthesis. GSS (LINF_250031200), GSS: glutathione synthetase (LINF_140015200), TRYS: trypanothione synthetase (LINF_270025600), glutathione peroxidase (LINF_360038100), trypanothione synthetase (LINF_230009800), spermidine synthase (LINF_040010800), trypanosome basal body component (LINF_240028000) and two mannosyltransferases (LINF_360027400 and LINF_120006700). S9C Fig) Genomic region showing the heterozygous mutation in GSS in two lines, AmBRcl.8 and AmBRcl.6 of *L. mexicana*. The reference genome and gene IDs were retrieved from TriTrypDB (https://tritrypdb.org). Images were produced using IGV 2.8.9 (http://software.broadinstitute.org).and STRING (https://string-db.org). See Fig 5 for a full description of the S9A and S9C Fig.
(TIF)

**S10 Fig. Summary of mutations in all polyene resistant lines of *Leishmania* spp.** (List of genes). Gene ID shown on the left were retrieved from TriTrypDB. All polyene resistant lines are shown in the x-axis and the mutation(s) type present in each are coloured coded. This image was produced with R package. See S9 Table for a full list of these variants.
(TIF)

**S11 Fig. Histopathology of primary lesions (footpad) of BALB/c mice inoculated with AmBR lines of *L. mexicana*.** S11A1-A2, S11B1-B2 Fig) AmBRcl.14 (yellow) and AmBRcl.3 (orange) infection with intense parasitic load and inflammatory infiltrate within skin histiocytes. S11C1-C2 Fig) *L. mexicana* wild type (green) causing diffuse inflammatory infiltration. Internalised amastigotes are indicated (black arrows). S11D1-D2, S11E1-E2 Fig) infection with AmBRcl.8 and AmBRcl.6 (dark and light blue) and discrete inflammatory reaction localised in papillary dermis without visible parasites. Samples are representative of three biological replicates. Hematoxylin-eosin (H&E) stain. Scale bar ~10 µM Objective ~63x.
(TIF)

**S12 Fig. Retention of resistance (EC50 values) to AmB in *L. mexicana*-AmBR lines recovered from mice tissue (footpad and lymph nodes) after treatment with AmB (1 mg/kg).** Susceptibility to AmB in AmBR *L. mexicana* axenic promastigotes after infection in mice. Parasites were recovered as amastigotes from mice tissue (lymph nodes and footpad) and transformed into promastigotes in HOMEM culture medium. Mice were treated with AmB (1 mg/kg) at week thirteen or left untreated (control group). See Material and Methods for a full description.
(TIF)

## Acknowledgments

The authors thank Glasgow Polyomics for WGS, GC-MS and LC-MS metabolomics data acquisition. This work made use of the facilities of the Hamilton HPC Service of Durham University for WGS data analyses.

## Author Contributions

**Conceptualization:** Edubiel A. Alpizar-Sosa, Michael P. Barrett.

**Formal analysis:** Edubiel A. Alpizar-Sosa, Nur Raihana Binti Ithnin, Wenbin Wei, Andrew W. Pountain, Stefan K. Weidt.

**Funding acquisition:** Edubiel A. Alpizar-Sosa, Nur Raihana Binti Ithnin, Paul W. Denny, Michael P. Barrett.

**Investigation:** Edubiel A. Alpizar-Sosa, Nur Raihana Binti Ithnin, Stefan K. Weidt, Anne M. Donachie, Ryan Ritchie.

**Methodology:** Edubiel A. Alpizar-Sosa, Nur Raihana Binti Ithnin, Andrew W. Pountain, Stefan K. Weidt, Anne M. Donachie, Ryan Ritchie.

**Supervision:** Richard J. S. Burchmore, Michael P. Barrett.

**Writing – original draft:** Edubiel A. Alpizar-Sosa, Michael P. Barrett.

**Writing – review & editing:** Edubiel A. Alpizar-Sosa, Andrew W. Pountain, Emily A. Dickie, Paul W. Denny, Michael P. Barrett.

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
