## [Decision Letter · Decision Letter 0]

29 May 2022

Dear Prof. Barrett,

Thank you very much for submitting your manuscript "Amphotericin B resistance in Leishmania mexicana: Alterations to sterol metabolism and oxidative stress response." for consideration at PLOS Neglected Tropical Diseases. As with all papers reviewed by the journal, your manuscript was reviewed by members of the editorial board and by several independent reviewers. In light of the reviews (below this email), we would like to invite the resubmission of a significantly-revised version that takes into account the reviewers' comments. 

1. The importance of hypersensitivity to Paromomycin should be included in Discussion.

2. As Leishmania species differ considerably in their biochemical responses and resistance patterns, could data with L. donovani be included as Ampho B is given in VL in SOuth Asia and the relevance of this work would increase manifold.

3. L.mexicana is treated with glucantime and not Ampho B, therefore how can this study be translated into management of L. Mexicana cutaneous Leishmaniasis?

We cannot make any decision about publication until we have seen the revised manuscript and your response to the reviewers' comments. Your revised manuscript is also likely to be sent to reviewers for further evaluation.

Sincerely,

Mitali Chatterjee

Associate Editor

Brian Weiss

Deputy Editor

1. The importance of hypersensitivity to Paromomycin should be included in Discussion.

2. As Leishmania species differ considerably in their biochemical responses and resistance patterns, could data with L. donovani be included as Ampho B is given in VL in SOuth Asia and the relevance of this work would increase manifold.

3. L.mexicana is treated with glucantime and not Ampho B, therefore how can this study be translated into management of L. Mexicana cutaneous Leishmaniasis?

Reviewer's Responses to Questions

**Key Review Criteria Required for Acceptance?**

**Methods**

-Are the objectives of the study clearly articulated with a clear testable hypothesis stated?

-Is the study design appropriate to address the stated objectives?

-Is the population clearly described and appropriate for the hypothesis being tested?

-Is the sample size sufficient to ensure adequate power to address the hypothesis being tested?

-Were correct statistical analysis used to support conclusions?

-Are there concerns about ethical or regulatory requirements being met?

Reviewer #1: Methods were fine

Reviewer #2: Yes methodology adopted is as per requirement of stated objectives

**Results**

-Does the analysis presented match the analysis plan?

-Are the results clearly and completely presented?

-Are the figures (Tables, Images) of sufficient quality for clarity?

Reviewer #1: see below

Reviewer #2: yes

**Conclusions**

-Are the conclusions supported by the data presented?

-Are the limitations of analysis clearly described?

-Do the authors discuss how these data can be helpful to advance our understanding of the topic under study?

-Is public health relevance addressed?

Reviewer #1: see below

Reviewer #2: No new knowledge genertated as already reported information is being retested.

**Editorial and Data Presentation Modifications?**

Reviewer #1: (No Response)

Reviewer #2: (No Response)

**Summary and General Comments**

Reviewer #1: Alpizar-Sosa et al. are presenting a very detailed and comprehensive study of in vitro AmB in the parasite Leishmania. Overall the work has been carried out expertly with many attentions to details. They are similarities with their previous study of Pountain et al., that was published in PNTD in 2019 but also a number of differences that bring new perspectives on the mode of action of AmB in Leishmania. Here for the first time they present data of cells selected for resistance to nystatin. Interestingly they have shown differences in AmB activity depending on the medium and furthered these studies with state of the art metabolomic work. Their collateral sensitivity to paromomycin, pentamidine and others is of interest and should also be emphasized. Since the genes discovered (C5DS, C24SMT, MT) are similar, and their surprising finding that resistant cells retained their virulence have already been reported in their 2019 paper I would suggest to the authors to emphasize how the current study distinguishes itself from the previous one. 

I am not sure I understand their statement on line 146-7. Why is their WT not growing when passing from HOMEM to DM?; did they not select WT cells for resistance when grown in DM? 

In conclusion a very complete and comprehensive study but I encourage to highlight more clearly how this study distinguishes from their previous one (outside studying 15 lines instead of 4). BTW their statement on line 512 is not accurate as other published studies have studied more than one AmB resistant mutant.

Reviewer #2: The manuscript entitled “ Amphotericin resistance in L. Mexicana : alteration to sterol metabolism and oxidative stress response” by Edubiel et al., demonstrates that in laboratory generated AmB resistant cell lines, wild type ergosterol is replaced by other sterol intermediates. Further, WGS of these cell lines revealed mutations in sterol methyl transferase and/or sterol C5 desaturase. In some cell lines, additional deletion of miltefosine transporter gene was also observed. However, most of the resistant cell lines (10/14) did not exhibit virulence both under in vitro and in vivo conditions suggesting loss of virulence.

The study did not generate any new information regarding mechanism of AmB resistance in laboratory generated resistant cell lines of L. mexicana ( as per ref nos. 93-104) and other Leishmania spp. Present manuscript is mere repetition of the same with some more cell lines. To understand whether, same deduced mechanism of AmB resistance in laboratory is indeed prevail in field, the study has to be performed with AmB resistant clinical isolats taking laboratory resistant cells as control. These clinical isolates should also be characterized in terms of in vivo resistant phenotype, virulence and cross resistance. Identified genes can be evaluated for their expression levels. Further to understand mechanism of cross resistance, comparative sterol profile and differential expression of sterol biosynthetic pathway may also be studied in mitefosine resistant lines. Hence,manuscript in present form is not recommended for publication.

PLOS authors have the option to publish the peer review history of their article (what does this mean?). If published, this will include your full peer review and any attached files.

Reviewer #1: No

Reviewer #2: No
---

## [Editor Report · Decision Letter 1]

31 Aug 2022

Dear Prof. Barrett,

We are pleased to inform you that your manuscript 'Amphotericin B resistance in Leishmania mexicana: Alterations to sterol metabolism and oxidative stress response.' has been provisionally accepted for publication in PLOS Neglected Tropical Diseases.

Best regards,

Mitali Chatterjee

Academic Editor

Brian Weiss

Section Editor

The revised manuscript has been extensively edited and has incorporated the suggestions of the reviewers.

---

## [Editor Report · Acceptance letter]

23 Sep 2022

Dear Prof. Barrett,

We are delighted to inform you that your manuscript, "Amphotericin B resistance in *Leishmania mexicana*: Alterations to sterol metabolism and oxidative stress response.," has been formally accepted for publication in PLOS Neglected Tropical Diseases.

Best regards,

Shaden Kamhawi

co-Editor-in-Chief

Paul Brindley

co-Editor-in-Chief
